# *O*-GlcNAcylation of MITF regulates its activity and CDK4/6 inhibitor resistance in breast cancer

Yi Zhang[1,7], Shuyan Zhou[1,7], Yan Kai[1,7], Ya-qin Zhang[2], Changmin Peng[1], Zhuqing Li[1], Muhammad Jameel mughal [1], Belmar Julie [3], Xiaoyan Zheng[4], Junfeng Ma [5], Cynthia X. Ma [6], Min Shen[2], Matthew D. Hall [2], Shunqiang Li[3] ✉ & Wenge Zhu[1] ✉

Cyclin-dependent kinases 4 and 6 (CDK4/6) play a pivotal role in cell cycle and cancer development. Targeting CDK4/6 has demonstrated promising effects against breast cancer. However, resistance to CDK4/6 inhibitors (CDK4/6i), such as palbociclib, remains a substantial challenge in clinical settings. Using high-throughput combinatorial drug screening and genomic sequencing, we find that the microphthalmia-associated transcription factor (MITF) is activated via O-GlcNAcylation by O-GlcNAc transferase (OGT) in palbociclib-resistant breast cancer cells and tumors. Mechanistically, O-GlcNAcylation of MITF at Serine 49 enhances its interaction with importin α/β, thus promoting its translocation to nuclei, where it suppresses palbociclib-induced senescence. Inhibition of MITF or its O-GlcNAcylation re-sensitizes resistant cells to palbociclib. Moreover, clinical studies confirm the activation of MITF in tumors from patients who are palbociclib-resistant or undergoing palbociclib treatment. Collectively, our studies shed light on the mechanism regulating palbociclib resistance and present clinical evidence for developing therapeutic approaches to treat CDK4/6i-resistant breast cancer patients.

The cyclin-dependent kinase (CDK) 4/6 are serine/threonine kinases that are associated with D-type cyclins, such as Cyclin D1, to promote the G1-S phase transition by phosphorylating retinoblastoma protein (RB), thus resulting in activation of E2F transcription factors that facilitate cell cycle progression[1–4]. CDK4/6 proteins are frequently upregulated and activated in a variety of cancers[5–8], making them attractive targets for therapeutic intervention. As a result, inhibiting CDK4/6 has emerged as a promising strategy for treating cancer. Presently, three CDK4/6 inhibitors (CDK4/6i), palbociclib, ribociclib, and abemaciclib, have gained FDA approval for the treatment of metastatic ER-positive, HER2-negative breast cancer in combination with hormone therapy[9–11]. Although the majority of patients initially respond well to CDK4/6i-based therapy and delay progression, the development of resistance poses a significant challenge to effective clinical management and long-term survival[12]. Although recent studies have implicated several resistant mechanisms, including mutations in RB1 or FAT1, and activation of CDK4/6, CDK2, PI3K/AKT, RAS/ERK, or STAT3[13–19], treatment strategies to overcome CDK4/6i resistance remain an unmet clinical need.

[1]Department of Biochemistry and Molecular Medicine, GWU Cancer Center, George Washington University School of Medicine and Health Sciences, Washington, DC, USA. [2]Division of Preclinical Innovation (Intramural), National Center for Advancing Translational Sciences (NCATS), National Institutes of Health, Rockville, MD, USA. [3]Department of Medicine, Washington University School of Medicine in St Louis, Siteman Cancer Center, St Louis, MO, USA. [4]Department of Anatomy and Cell Biology, GWU Cancer Center, George Washington University School of Medicine and Health Sciences, Washington, DC, USA. [5]Department of Oncology, Lombardi Comprehensive Cancer Center, Georgetown University Medical Center, Washington, DC, USA. [6]Division of Oncology, Department of Internal Medicine, Washington University School of Medicine, St. Louis, MO, USA. [7]These authors contributed equally: Yi Zhang, Shuyan Zhou, Yan Kai. ✉e-mail: shunqiangli@wustl.edu; wz6812@gwu.edu

Microphthalmia-associated transcription factor (MITF) belongs to the basic-helix-loop-helix-leucine zipper (bHLH-ZIP) family, constituting the MiT/TFE transcription factor family in conjunction with TFEB, TFEC, and TFE3[20]. These MiT/TFE members regulate downstream gene expression by binding to the E-box motif, characterized by the palindromic canonical sequence of CACGTG[21]. MITF encompasses several isoforms with distinct amino termini, including MITF-A, MITF-C, MITF-E/D, MITF-B, MITF-C, MITF-H, and MITF-M[22]. Unlike other MITF isoforms, MITF-M is the predominant isoform expressed in melanocytes and melanoma cells, making it a potential biomarker of melanoma[23]. Under typical growth conditions, MITF-M is constitutively retained within nuclei[24]. In contrast, MITF-A, along with other isoforms, features an N-terminal 1B1b domain with specific residues facilitating their cytoplasmic localization. This localization arises due to cytoplasmic retention after phosphorylation of S173 by TAK1 in osteoclasts[25], or via phosphorylation by mTORC1 and the interaction with the RAG GTPases at the surface of the lysosome[26]. Notably, exogenous expression of MITF-A led to predominantly cytoplasmic localization, while mutation of conserved residues Q62 and L63 within exon 1B1b prevents cytoplasmic retention of MITF-A[27]. The distinct cellular localization of MITF-A implies potential divergent functionality compared to MITF-M in cells. So far, compared to MITF-M, our understanding of the regulatory mechanism governing MITF-A in cancer cells remains limited[23]. Even less is known about the physiological roles of MITF-A and other isoforms in response to drug treatments and drug resistance.

O-GlcNAcylation is a unique type of posttranslational modification that involves the addition of a single O-link N-acetylglucosamine (O-GlcNAc) sugar moiety to the serine (Ser) or threonine (Thr) residues of target proteins by O-GlcNAc transferase (OGT) and removed by O-GlcNAase (OGA)[28–30]. Dysregulation of O-GlcNAcylation has been linked to various diseases, such as diabetes, neurodegenerative disorders, and cancers[31–34]. Emerging evidence has indicated that aberrant O-GlcNAcylation may contribute to tumorigenesis, promote drug resistance, and influence the immune response by altering protein stability, localization, or interactions[35–38]. However, the role of O-GlcNAc signaling in the regulation of CDK4/6i resistance remains unclear.

In this study, we perform an unbiased quantitative high-throughput combinatorial screening (qHTCS) to identify compounds capable of overcoming resistance to CDK4/6i in breast cancer cells. Through this screening, we identify ML329, an MITF inhibitor, which effectively restores the sensitivity to palbociclib in CDK4/6i-resistant cells, both in vitro and in vivo. In-depth mechanistic analyses reveal that palbociclib-resistant breast cancer cells exhibit elevated levels of MITF-A and its O-GlcNAcylation. Notably, O-GlcNAcylation of MITF-A at Serine 49 (S49) by O-GlcNAc transferase (OGT) facilitates its interaction with importin α/β, thereby facilitating nuclear translocation. Within the nucleus, MITF-A curtails palbociclib-induced senescence, ultimately driving resistance. Inhibition of MITF or its O-GlcNAcylation sensitizes resistant cells to palbociclib both in vitro and in vivo. This previously unappreciated O-GlcNAcylation-MITF-mediate mechanism is corroborated in resistant patient-derived xenograft (PDX) lines, as well as tumor samples from patients who receive CDK4/6i treatment. Collectively, these findings not only shed light on an innovative regulatory mechanism governing MITF's activity and its role in palbociclib resistance but also provide a potential therapeutic strategy for effectively managing palbociclib-resistant breast cancer patients.

## Results
### qHTCS identifies MITF inhibitor ML329 that overcomes palbociclib resistance
To identify the effective therapeutic strategies to overcome palbociclib-resistance in breast cancer cells, we generated two palbociclib-resistant (PR) breast cancer cells (MCF-7 PR and T-47D PR) by continuously treating cells with palbociclib using approaches as we

described previously[39] (Supplementary Fig. 1a, b). The MCF-7 PR cells also exhibited resistance to the other two CDK4/6i, ribociclib and abemaciclib (Supplementary Fig. 1c, d). Using MCF-7 and MCF-7 PR cells, we conducted a two-round qHTCS with MIPE (Mechanism Interrogation Plate) and NPC (NCATS Pharmaceutical Collection) small molecular compound libraries. From the first-round screening, we identified a total of 120 compounds that efficiently suppressed the proliferation of both MCF-7 and MCF-7 PR cells. To identify compounds that synergize with palbociclib in MCF-7 PR cells, we selected the 20 most active compounds against MCF-7 PR cells for the second-round screen together with palbociclib (Fig. 1a). In the second screening, a MITF inhibitor called ML329 was found to be one of the top hits that could resensitize MCF-7 PR cells to palbociclib (Fig. 1a). To validate the synergistic effects of ML329 with palbociclib, MCF-7 PR and T-47D PR cells were treated with a combination of ML329 and palbociclib for 72 h and cell viability was measured and calculated using Combenefit as described[40]. Significantly, the combined treatment of ML329 with palbociclib inhibited the growth of both resistant cells (Fig. 1b). To precisely measure the synergy, we calculated the combination index (CI), a quantitative measure of the interaction between two drugs (synergism: CI < 1; additive effect: CI = 1; and antagonism: CI > 1)[41]. The combination of palbociclib and ML329 exhibited a great synergy in both resistant cells (Supplementary Fig. 1e, f).

To test whether combinatorial treatment affects cell cycle progression, we examined the proliferation of resistant cells by using the Edu incorporation assay. Palbociclib or ML329 alone had little or limited impact on cell cycle progression as indicated by S phase population of MCF-7 PR cells, however, the combinatorial treatment of ML329 and palbociclib dramatically reduced the S phase population of MCF-7 PR cells, suggesting that combination but not individual drug treatments lead to a cell cycle arrest before S phase (Fig. 1c). In agreement with these synergistic effects, the combinatorial treatment led to a substantial reduction in the colony formation capacity of both resistant cells (Fig. 1d, e) and re-sensitized resistant MCF-7 PR tumors to palbociclib in vivo (Fig. 1f, g and Supplementary Fig. 1g). Collectively, these results suggest that MITF inhibitor ML329 is able to overcome palbociclib resistance in breast cancer cells.

### Inhibition of MITF overcomes palbociclib resistance by activating the senescence pathway in breast cancer cells
We next investigated how MITF regulates palbociclib resistance. Given that the MITF gene transcribes multiple isoforms[42], we first set to determine which MITF isoform is enriched in resistant cells. Using RT-PCR, we found that MITF-A is the prevalent isoform highly expressed in both MCF-7 PR and T-47D PR cells compared to their sensitive counterparts (Supplementary Fig. 2a) (We refer MITF-A as MITF in the following text). Both RNA-seq and Western blot analyses indicated that MITF mRNA and protein expressions were significantly increased in resistant cells compared to sensitive cells (Fig. 2a, b). Interestingly, suppression of MITF by small hairpin RNA (shRNA) re-sensitized both resistant cells to palbociclib as indicated by cell survival and colony formation assays (Fig. 2c and Supplementary Fig. 2b–d). Consistently, depletion of MITF by shRNA together with palbociclib treatment resulted in a significant decrease of phosphorylated Rb, while simultaneously leading to the increase of p21 level compared to treatment with palbociclib alone (Fig. 2d and Supplementary Fig. 2e), indicating that MITF depletion together with palbociclib leads to cell cycle arrest of resistant cells. Similar results were observed when MITF was inhibited by ML329 treatment in both resistant cells (Supplementary Fig. 2f). In agreement with these in vitro results, inhibition of MITF by shRNA re-sensitized MCF-7 PR tumors to palbociclib in vivo (Fig. 2e, f and Supplementary Fig. 2g). Thus, MITF appears to play an important role in regulating palbociclib resistance in CDK4/6i resistant breast cancer cells.

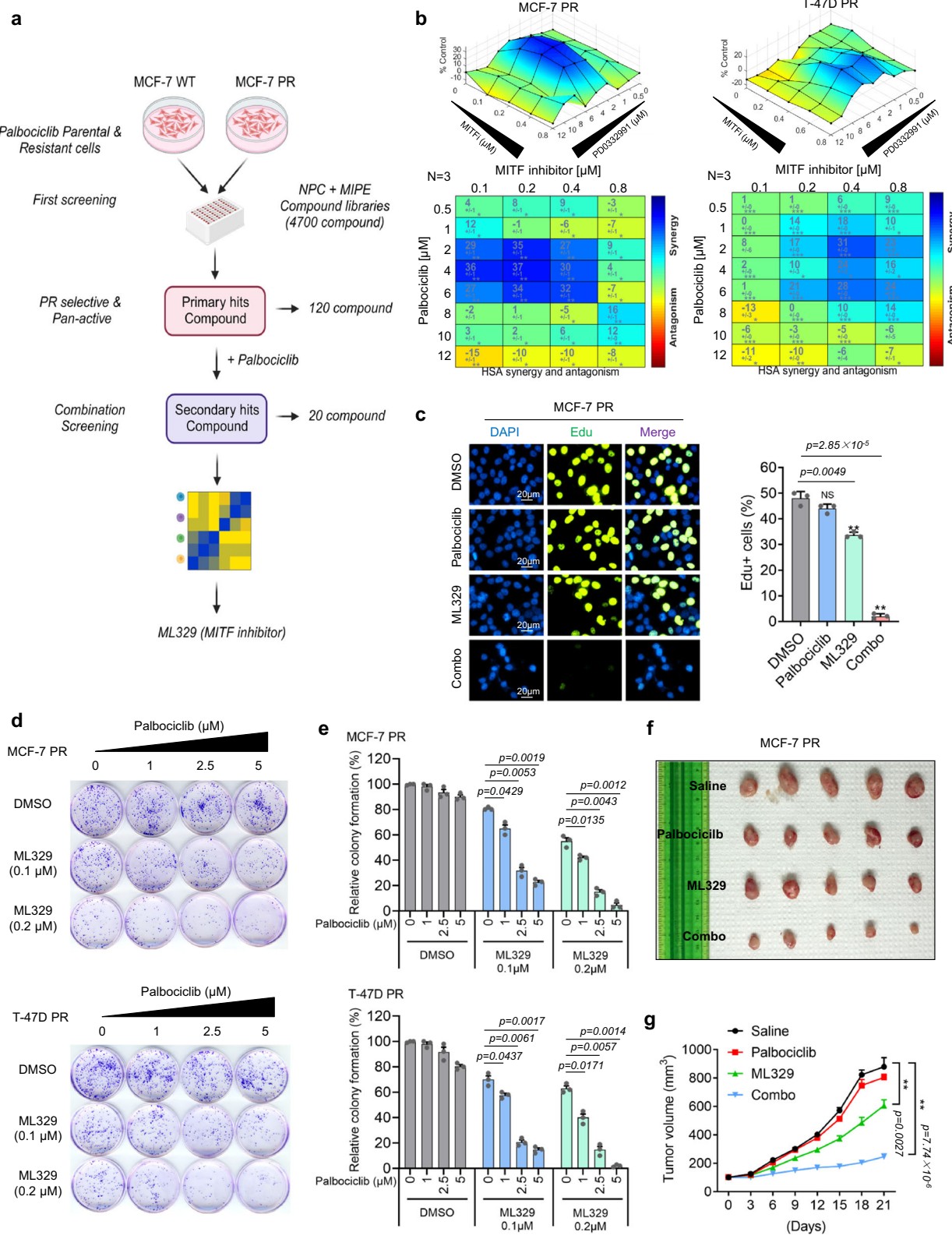

**Fig. 1 | qHTCS identifies MITF inhibitor ML329 that overcomes palbociclib resistance. a** Schematic of qHTCS to identify ML329 that can overcome palbociclib resistance. **b** Surface plots (top) and synergy matrix (bottom) to exhibit the synergy between palbociclib and ML329 in MCF-7 PR and T-47D PR cells (*n* = 3 independent experiments). **c** Edu staining to measure the S phase population in MCF-7 PR cells treated as indicated. The scale bar represents 20 μm. (*n* = 3 independent experiments). **d, e** Representative images of colony formation (**d**) and quantification results (**e**) in MCF-7 PR and T-47D PR cells with indicated treatments (*n* = 3 independent experiments). **f, g** Representative images (**f**) and growth curves (**g**) of MCF-7 PR xenograft tumors treated with saline, palbociclib (25 mg/kg), ML329(5 mg/kg), or a combination of both for 3 weeks. *n* = 6 mice/group. **\**p* ≤ 0.01. All error bars are expressed as mean ± SEM. Two-tailed Student's *t* tests were employed for statistical evaluation. Source data are provided as a Source Data file.

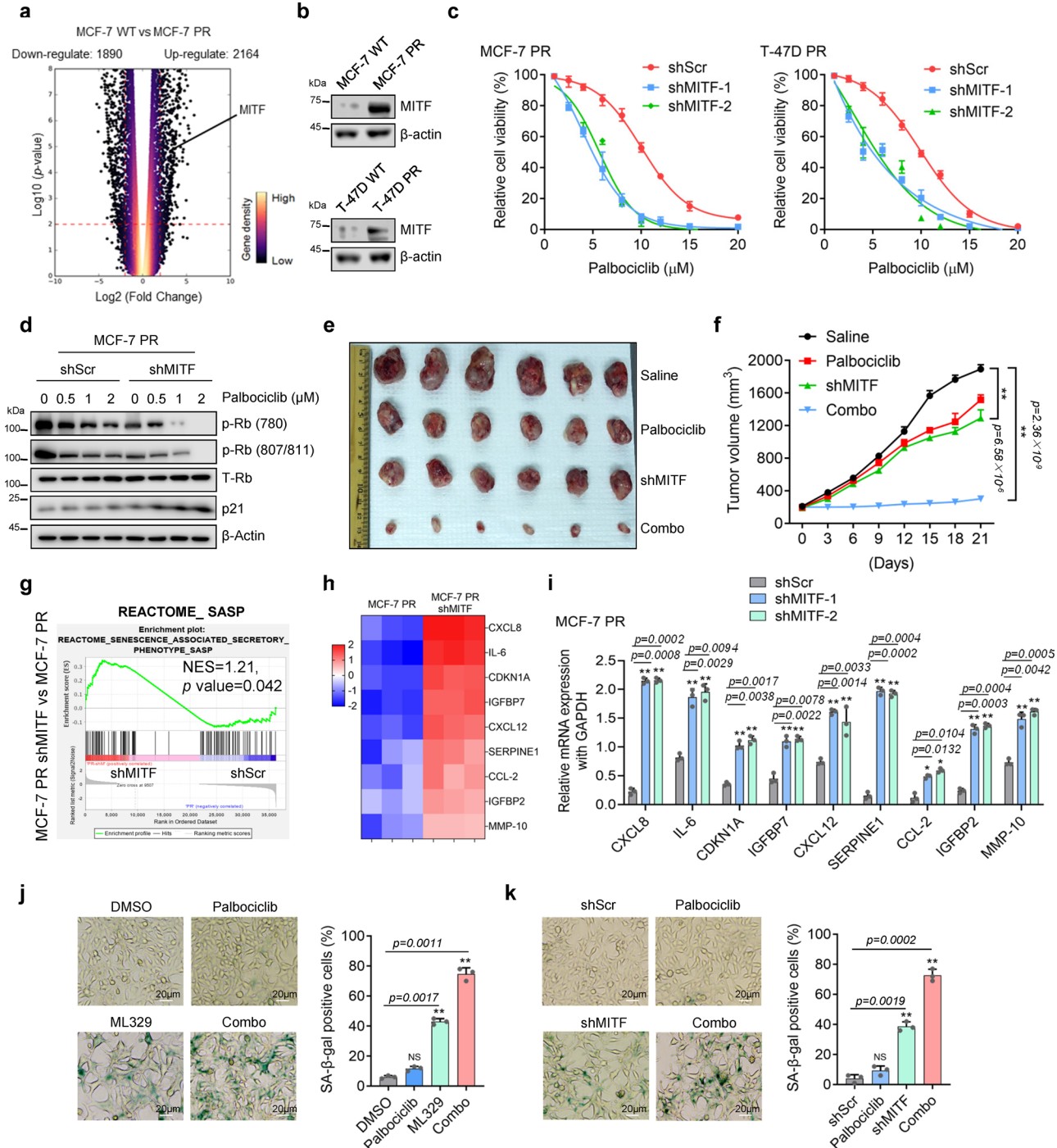

**Fig. 2 | Inhibition of MITF overcomes palbociclib resistance by activating the senescence pathway in breast cancer cells. a** Volcano plot to display upregulated and downregulated genes in MCF-7 PR cells compared to MCF-7 cells. A change is considered significant if the *p-value* is <0.05 and the change is >2-fold. **b** Cell lysates from MCF-7 and MCF-7 PR cells were immunoblotted for indicated proteins. (*n* = 3 independent experiments). **c** Cell viability was examined in MCF-7 PR and T-47D PR cells treated with increasing concentration of palbociclib for 3 days after MITF depletion by shRNA (*n* = 3 independent experiments). **d** MCF-7 PR cells transfected with shScr or shMITF were treated with palbociclib, followed by immunoblotting for indicated proteins. (*n* = 3 independent experiments). **e, f** Representative images (**e**) and growth curves (**f**) of MCF-7 PR xenograft tumors with indicated treatment for 3 weeks. *n* = 6 mice/group. **g** GSEA profiling to show the enrichment of the SASP geneset in MCF7 PR cells treated with shScr or shMITF. NES score and p values were determined by GSEA software. **h** Heatmap profiling of SASP-associated factors in MCF-7 PR cell treated with shScr or shMITF. The displayed SASP-associated factors were selected from the top list of genes based on expression fold change. **i** MCF7 PR cells treated as indicated were harvested and then subjected to qPCR to examine the expression of SASP-associated factors shown in **h** (*n* = 3 independent experiments). **j, k** Representative images of SA-β-gal staining in MCF-7 PR cells treated with ML-329 (**j**) or shRNAs (**k**) (*n* = 3 independent experiments). The scale bar represents 20 μm. **p ≤ 0.01, *p ≤ 0.05. All error bars are expressed as mean ± SEM. Two-tailed Student's *t* tests were employed for statistical evaluation. Source data are provided as a Source Data file.

Since MITF inhibition together with palbociclib increased expression of p21 (Fig. 2d), a marker of senescence[43], we hypothesized that depletion of MITF re-sensitizes PR cells to palbociclib by inducing senescence. Indeed, inhibition of MITF resulted in a significant enrichment of the senescence-associated secretory phenotype (SASP) geneset in MCF-7 PR cells (Fig. 2g). Meanwhile, RNA-seq analysis indicated that the expressions of multiple SASP genes (CXCL8, IL-6, CDKN1A, IGFBP7, CXCL12, SERPINE1, CCL-2, IGFBP2, and MMP-10) were highly induced in MITF-depleted cells (Fig. 2h). These up-regulated genes upon MITF depletion were further confirmed by qPCR analysis (Fig. 2i). Consistently, inhibition of MITF alone increased the number of cells with positive staining of senescence-associated β-galactosidase (SA-β-Gal); and inhibition of MITF by siRNA or ML329 in combination with palbociclib further augmented the cell population undergoing senescence in MCF-7 PR cells (Fig. 2j, k). Additionally, co-administration of ML329 and palbociclib did not induce apoptosis as indicated by the absence of cleaved PARP (Supplementary Fig. 2h). Thus, MITF inhibition overcomes palbociclib resistance by activating senescence in resistant breast cancer cells.

### OGT interacts with MITF and promotes its nuclear translocation

To determine how MITF regulates palbociclib resistance, we first examined the subcellular localization of MITF in sensitive and resistant cells. Surprisingly, unlike sensitive cells in which MITF predominantly resided in the cytoplasm, MITF primarily localized in nuclei in both resistant cells (Fig. 3a). Previous studies reported that MITF is phosphorylated by mTORC1 and this phosphorylation leads to its interaction with 14-3-3 proteins and cytoplasmic retention[26,44]. In agreement with this observation, the interaction of MITF with 14-3-3 was decreased in both resistant cells compared to their sensitive counterparts (Supplementary Fig. 3a, b). Additionally, the association between MITF and 14-3-3 was diminished upon Torin1 (a mTORC1 inhibitor) treatment (Supplementary Fig. 3a, b). Thus, MITF primarily localizes within nuclei in resistant cells compared to sensitive cells.

To explore the mechanism that supersedes cytoplasmic retention of MITF in resistant cells, we conducted a mass spectrometry analysis to identify MITF-associated proteins in MCF-7 PR cells. Notably, we found that OGT, the O-linked N-acetylglucosamine (GlcNAc) transferase, was significantly enriched in MITF immunoprecipitated fraction (Fig. 3b). Co-immunoprecipitation (Co-IP) assay validated the association between endogenous MITF and OGT, and vice versa (Fig. 3c). Consistently, OGT directly O-GlcNAcylated MITF in vitro (Fig. 3d). Given that OGT was highly expressed in resistant cells compared to sensitive cells (Supplementary Fig. 3c), we postulated that OGT might regulate MITF activity through O-GlcNAcylation in resistant cells. Indeed, overexpression or knockdown of OGT significantly increased or decreased O-GlcNAcylation of MITF respectively (Fig. 3e, f), and inhibition of OGT did not affect MITF protein levels but significantly reduced its nuclear accumulation in MCF-7 PR cells (Fig. 3g and Supplementary Fig. 3d, e). In agreement with these data, OGT depletion or treatment with the OGT inhibitor OSMI-1 significantly increased the interaction between MITF and 14-3-3 (Fig. 3h, i), whereas this interaction was reduced upon PUGNAc (an OGA inhibitor) treatment (Fig. 3j). Moreover, overexpression of OGT promoted the translocation of MITF from the cytoplasm into the nucleus in sensitive MCF-7 cells (Supplementary Fig. 3f). Furthermore, we synthesized an oligonucleotide containing the MITF binding motif (E-box), and tagged it with biotin, and then mixed it with nuclear lysates from cells. The results indicated that OGT depletion significantly decreased the interaction of nuclear MITF with E-box DNA oligos (Fig. 3k). Accordingly, depletion of OGT also sensitized both resistant cells to palbociclib (Fig. 3l, m). Collectively, the above results strongly indicate that OGT directly interacts with and O-GlcNAcylates MITF, which in turn prevents its interaction with 14-3-3, promoting its nuclear translocation.

### O-GlcNAcylation of MITF at S49 is required for nuclear accumulation and resistance

To further investigate how O-GlcNAcylation regulates MITF activity, we conducted a mass spectrometry assay to identify the specific O-Glycosylation site of MITF. The analysis found that O-GlcNAcylation of MITF predominantly occurred on a peptide that spans residues 34-56, and S49 was the most likely O-Glycosylation site (Fig. 4a). To exclude the possibility of O-GlcNAc modifications on the MITF peptide was resulted from other Ser/Thr sites rather than S49 site, we mutated S49 as well as other potential O-GlcNAcylation sites from serine to alanine. Interestingly, the O-GlcNAc level of MITF was markedly reduced only in S49A mutant in vitro and in vivo (Fig. 4b, c), suggesting that S49 is the primary O-GlcNAc site of MITF. Importantly, sequence comparison analysis showed that MITF S49 was conserved across various species (Supplementary Fig. 4a).

We next investigated how S49 O-GlcNAcylation regulates MITF activity. Interestingly, the mutant MITF (S49A) exhibited reduced levels of O-GlcNAc and increased interaction with 14-3-3 compared to wild-type (WT) MITF (Fig. 4d). Moreover, both nuclear fractionation and IF assays indicated that WT MITF but not MITF(S49A) displayed an increased nuclear localization when O-GlcNAcylation was enhanced by OGT overexpression (Fig. 4e, f). Consistently, WT MITF mainly localized within the nucleus, therefore resulting in an increased interaction of nuclear MITF with E-box DNA compared to S49A mutant (Fig. 4g). Depletion of MITF in MCF-7 PR cells reduced the expression of MITF targeting genes CCND1, BIRC1, and CCNB1, which was restored by the expression of WT MITF but not the S49A mutant (Fig. 4h). The above results indicated a critical role of O-GlcNAcylation of MITF at S49 in the regulation of its nuclear localization and transcriptional activity.

We next investigated how O-GlcNAcylation at S49 regulates the response of resistant cells to palbociclib. Interestingly, depletion of MITF re-sensitized MCF-7 PR cells to palbociclib, which was rescued by reconstitution of WT MITF but not S49A mutant (Fig. 4i and Supplementary Fig. 4b). Using xenografted tumor models, we also found that reintroduction of WT MITF but not S49A mutant in MITF-depleted-MCF-7 PR cells restored resistance to palbociclib. Thus, O-GlcNAcylation at S49 of MITF plays a critical role in the regulation of palbociclib resistance in breast cancer cells.

### O-GlcNAcylation within the nuclear localization signal promotes the interaction of MITF with importin α/β

We next explored how O-GlcNAcylation regulates the nuclear translocation of MITF in resistant cells. Our previous study indicated that importin α functions as a reader of O-GlcNAcylated NLSs on cargo proteins, thus facilitating the nuclear translocation of these cargo proteins[45]. Given that MITF-A localizes in cytoplasm and features an N-terminal 1B1b domain that is absent in MITF-M, we anticipated that the 1B1b domain may play a key role in regulating its cellular localization. Using the NLS prediction program[46], we found that MITF contains two putative nuclear localization signal (NLS) regions within its N-terminus (Fig. 5a). Depletion of the first NLS, but not the second NLS, hindered the nuclear accumulation of MITF in MCF-7 PR cells (Fig. 5b). Consistently, addition of this NLS to N-terminus of eIF2B$\varepsilon$, a cytoplasmic protein[47,48], resulted in its re-localization to the nucleus (Supplementary Fig. 5a). Moreover, the NLS-eIF2B$\varepsilon$ exhibited an increased interaction with importinα4 and β compared to the eIF2B$\varepsilon$ (Supplementary Fig. 5b).

Since O-GlcNAcylation site S49 is localized within the first NLS, we assumed that O-GlcNAcylation of MITF regulates its nuclear translocation by affecting its interaction with importin α/β, which are membrane-associated proteins playing a critical role in regulating the transport of proteins between cytosol and nuclei[49]. To identify which importin is involved in the regulation of MITF nuclear translocation, we examined MITF localization by IF in cells with depletion of various importins using siRNAs. Depletion of importin α4, α7, and α8 appeared to compromise MITF nuclear localization and the

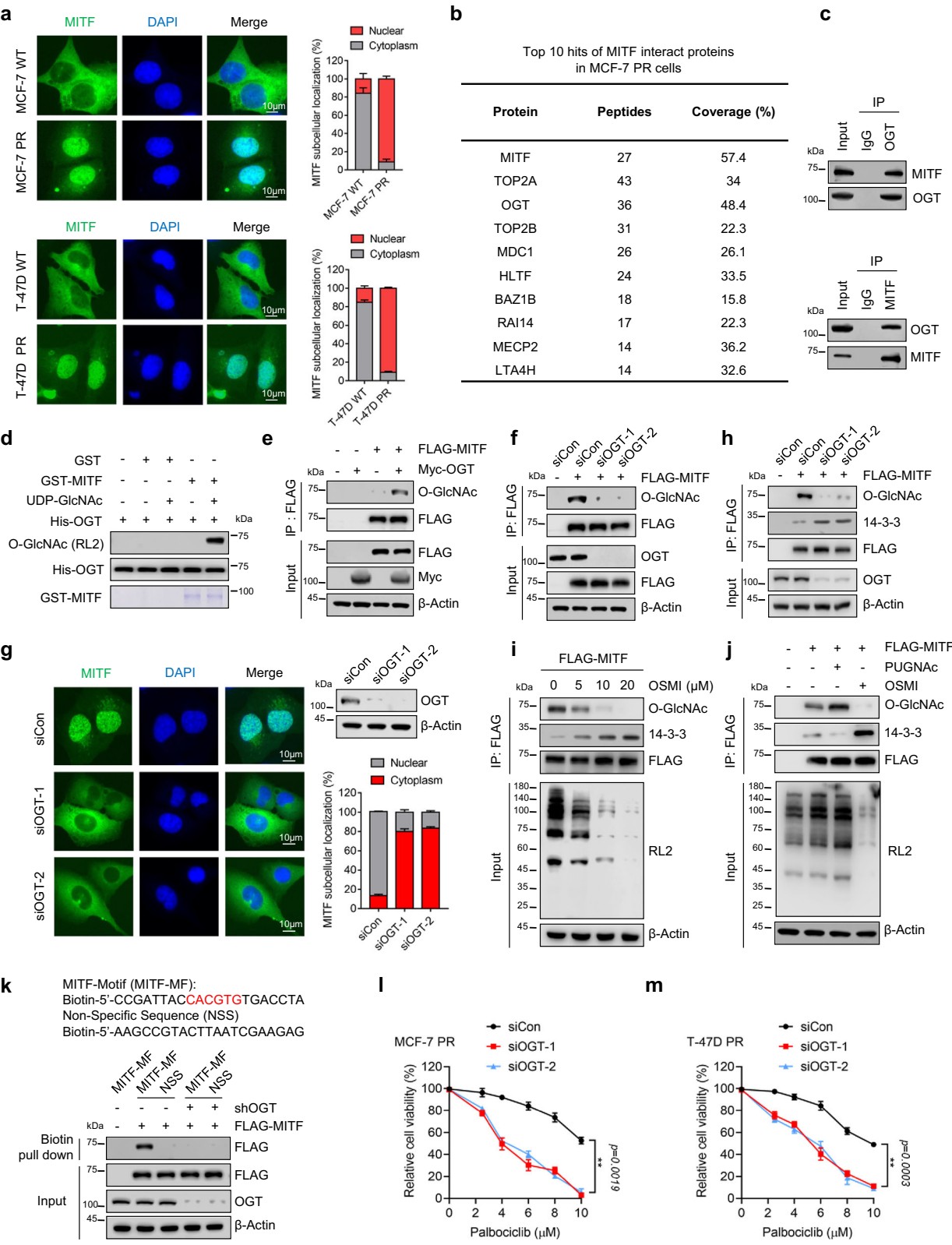

knockdown of these three importin proteins together led to the robust decrease of nuclear translocation of MITF (Fig. 5c and Supplementary Fig. 5c), suggesting that importin α4, α7, and α8 are essential for MITF nuclear translocation. Moreover, we also observed that suppression of importin β reduced the nuclear translocation of MITF (Supplementary Fig. 5d). These findings prompted us to test whether O-GlcNAcylation regulates the nuclear translocation of MITF by influencing the association of MITF with importin α/β. Intriguingly, inhibition of OGT in

MCF-7 PR cells dramatically reduced the interactions of MITF with importins α4, α7, α8, as well as importin β (Fig. 5d, e).

To further study how O-GlcNAcylation regulates the interaction between MITF and these importins, we generated O-GlcNAcylated MITF by in vitro O-GlcNAcylation assay. As shown in Fig. 5f, O-GlcNAcylated MITF displayed an enhanced interaction with these importins compared to non-O-GlcNAcylated MITF by GST pull-down assay, suggesting that O-GlcNAcylation promotes the recognition of

**Fig. 3 | OGT interacts with MITF and promotes its nuclear translocation.**
**a** Immunofluorescence to examine the MITF localization in cells as indicated (*n* = 3 independent experiments). The scale bar represents 10 µm. Right low, quantification of data shown on the left. **b** Mass spectrometry analysis to identify MITF-associated proteins in MCF-7 PR cells. **c** Co-immunoprecipitation (co-IP) to detect the interaction of indicated proteins in MCF-7 PR cells. (*n* = 3 independent experiments). **d** In vitro O-GlcNAcylation assay to examine O-GlcNAcylation of MITF. Recombinant GST-MITF and His-OGT were purified from *E. coli*. (*n* = 3 independent experiments). **e** HEK293T cells were transfected with the indicated plasmids for 48 h before being harvested for co-IP assay. FLAG-IPs were immunoblotted for indicated proteins. (*n* = 3 independent experiments). **f** HEK293T cells were transfected with the indicated siRNAs and plasmids for 48 h before being harvested. FLAG-IPs were immunoblotted for the indicated proteins. (*n* = 3 independent experiments). **g** MCF-7 PR cells were transfected with the indicated siRNAs for 48 h, followed by immunostaining to examine the localization of MITF (*n* = 3 independent experiments). The scale bar represents 10 µm. Right up, cells shown in the left image were harvested and immunoblotted for indicated proteins; Right low, quantification of data shown in the left. **h** MCF-7 PR cells were transfected with the indicated siRNAs and plasmids for 48 h before being harvested. FLAG-IPs were immunoblotted for the indicated proteins. (*n* = 3 independent experiments). **i, j** MCF-7 PR cells were treated as indicated for 48 h and then harvested for co-IP. FLAG-IPs were immunoblotted for indicated proteins. (*n* = 3 independent experiments). **k** MCF-7 PR cells were transfected with indicated plasmids, and the nuclear fractions were subjected to the biotin pull-down assay, followed by immunoblotting for indicated proteins. (*n* = 3 independent experiments). Upper, sequence of MITF-DNA-binding motif tagged with biotin. The red color represents the E-box of the MITF binding motif. **l, m** Cell viability of MCF-7 PR cells (**l**) and T-47D PR cells (**m**) treated with increasing concentration of palbociclib for 3 days after MITF depletion by siRNA (*n* = 3 independent experiments). **\*\***$p \le 0.01$. All error bars are expressed as mean ± SEM. Two-tailed Student's *t* tests were employed for statistical evaluation. Source data are provided as a Source Data file.

the NLS of MITF by these importins. Moreover, WT MITF exhibited a stronger interaction with importins α4, α7, and α8 than S49A mutant, and increasing O-GlcNAcylation by PUGNAc treatment further enhanced the association of WT MITF but not S49A mutant with importins α4, α7, and α8 (Fig. 5g–i). The aforementioned results strongly suggest that importin α/β functions as a sensor to recognize the O-GlcNAcylated NLS of MITF and facilitate its nuclear translocation.

### O-GlcNAcylation of MITF contributes to palbociclib resistance by regulating senescence signaling

To investigate how MITF regulates palbociclib resistance in breast cancer cells, we performed MITF ChIP-seq in MCF-7 and MCF-7 PR cells. Interestingly, the ChIP-seq identified over 3000 MITF-occupied sites located in both inter and intragenic regions throughout the genome in MCF-7 PR cells (Fig. 6a). In comparison, MITF exhibited a relatively higher binding intensity in MCF-7 PR cells compared to sensitive cells (Fig. 6b). From ChIP-seq analysis, we observed that MITF displayed a stronger affinity towards the promoter region of four SASP factors, SERPINE1, IL-6, CSF1, and PTGER2 in MCF-7 PR cells compared to MCF-7 cells (Fig. 6c). All these four genes have been reported to positively regulate senescence individually[50–52]. We assumed that MITF may suppress the expression of these genes in resistant cells. Indeed, ChIP-qPCR indicated that MITF specifically bound to the promoter region of these four genes (TYR gene was used as a positive control) (Fig. 6d), and depletion of MITF increased the expression of these genes (Fig. 6e, f). These results suggest that MITF directly represses the expression of these SASP genes, which in turn leads to resistance by suppressing senescence in resistant cells.

To determine whether O-GlcNAcylation of MITF regulates the expression of these SASP genes, we reconstituted MITF-depleted cells with WT MITF and S49A mutants, followed by qPCR analyses. Intriguingly, transcriptional inhibition of these SASP genes was restored only upon the reexpression of WT MITF, but not the S49A mutant, in resistant cells with endogenous MITF depletion (Fig. 6g, h). Consistently, only ectopic expression of WT MITF was able to overcome senescence induced by MITF knockdown, as evidenced by reduced p21 expression and SA-β-Gal staining, whereas such an effect was not observed in cells expressing the S49A mutant (Fig. 6i, j). These data collectively indicate that O-GlcNAcylation of MITF at S49 is essential for its activity in the regulation of senescence-associated secretome in palbociclib-resistant cells.

### MITF is activated in response to palbociclib and elevated in tumors from palbociclib-resistant breast cancer patients

We next conducted clinical studies to investigate whether or not elevated OGT-MITF axis is seen in patients who are undergoing CDK4/6i treatment or have developed resistance to CDK4/6i. Given that

increased MITF levels contribute to palbociclib resistance, it is possible that palbociclib treatment may result in increased expression of MITF as a compensatory survival mechanism. Indeed, both MITF mRNA and protein expression levels were increased upon palbociclib treatment in MCF7 cells (Fig. 7a). The similar pattern was also observed in cells in response to the treatment of another CDK4/6 inhibitor ribociclib (Supplementary Fig. 6a). We next employed an ER-positive breast cancer patient-derived xenograft (PDX) model to evaluate the effects of prolonged palbociclib treatment. Of note, the administration of palbociclib treatment to PDX tumor-bearing mice also led to a substantial increase in the expression level of MITF compared to the group without receiving the drug (Supplementary Fig. 6b). Thus, both in vitro and in vivo studies indicate that palbociclib treatment can stimulate MITF expression.

These data encouraged us to test whether the MITF is also activated in patients receiving palbociclib treatment. To this end, we analyzed the gene expression data of patient tumors obtained from the NeoPalAna clinical trial, which included estrogen receptor-positive breast cancer patients undergoing tumor biopsies before and after palbociclib treatment[53]. In line with the aforementioned findings in the preclinical setting, palbociclib treatment significantly increased MITF mRNA expression 16-20 weeks after palbociclib administration (surgery stage) (Fig. 7b and Supplementary Fig. 6c). Meanwhile, MITF expression was also significantly elevated at C1D15 stage in the palbociclib-resistant patients compared to sensitive patients (Fig. 7c). To further explore the role of MITF in the regulation of palbociclib resistance in breast cancer, we created a MITF signature geneset by overlapping proteins that interact with MITF (mass-spec analysis) and genes that are known to be associated with MITF function (PahtwayNet analysis) (Supplementary Fig. 6d). Strikingly, the MITF signature geneset was significantly enriched in patients after palbociclib treatment compared to pre-treatment (Fig. 7d and Supplementary Fig. 6e). Consistently, the same signature geneset was also elevated in the MCF-7 PR cells compared with its sensitive counterpart MCF-7 cells (Supplementary Fig. 6f).

To further validate whether increased MITF levels are a critical factor regulating palbociclib resistance in breast cancer, we examined the expression of MITF in a group of PDX lines, which consisted of 7 palbociclib-sensitive samples and 7 palbociclib-resistant samples. Significantly, expression levels of MITF, OGT, and O-GlcNAc were increased in most palbociclib-resistant PDX lines compared to sensitive lines (Fig. 7e and Supplementary Fig. 6g). Of note, both the MITF expression level and the O-GlcNAc expression levels were increased in the same PDX tumors before (ID# WHIM20, sensitive) and after (ID# WHIM20AR, resistance) the development of resistance to palbociclib due to long-term treatment with palbociclib in vivo, and MITF was predominantly located in the nuclei in resistant tumors (Supplementary Fig. 6h-i). Breast cancer organoids

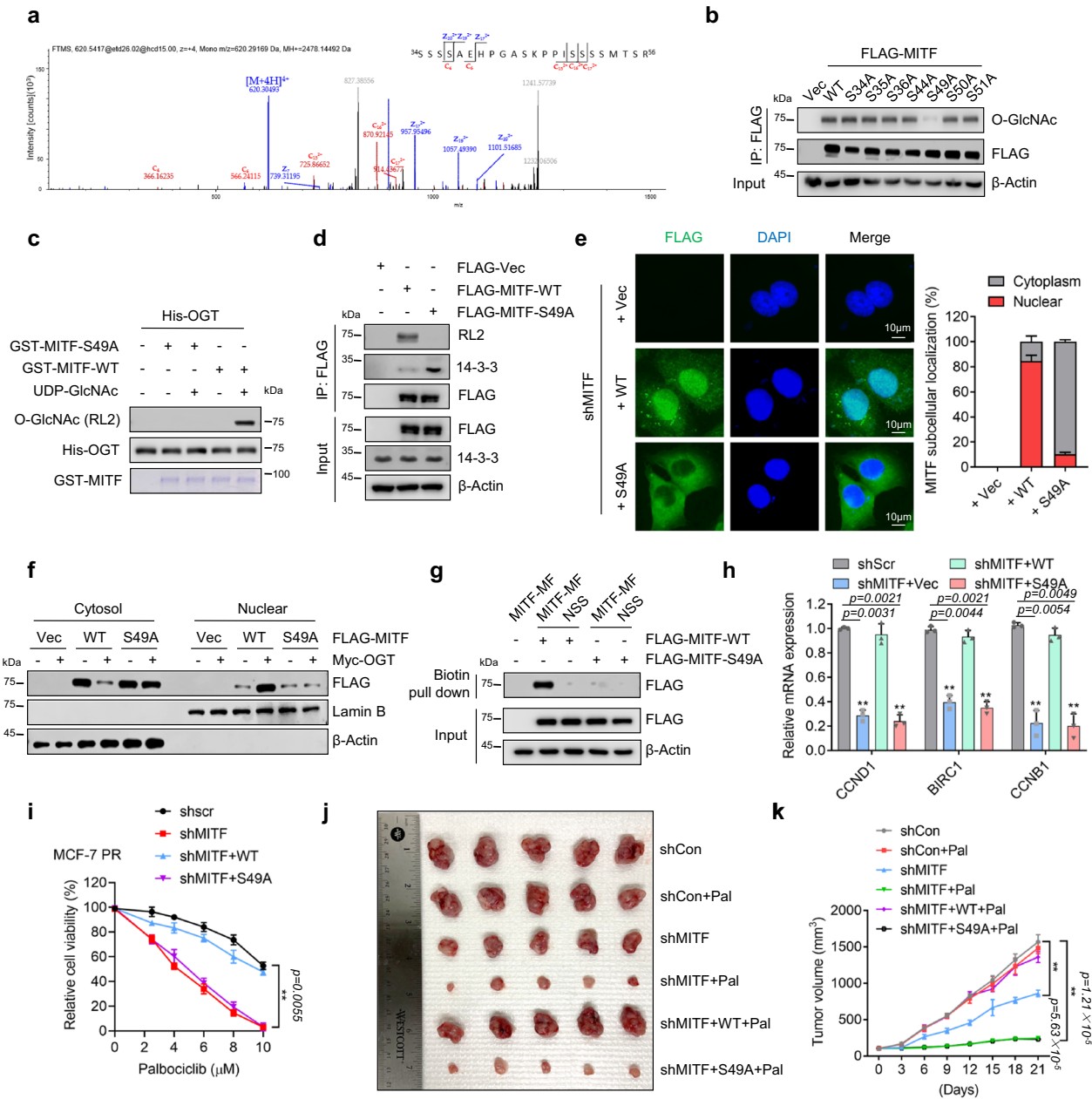

**Fig. 4 | O-GlcNAcylation of MITF at S49 is required for its nuclear accumulation.** **a** MITF was purified from HEK293T cells and analyzed by LC-MS/MS analysis to identify the potential O-GlcNAcylation sites. The peptide of MITF from amino acids 34–56 with O-GlcNAcylation identified by LC-MS/MS was shown (note: S49 is a potential O-GlcNAc site). **b** HEK293T cells expressing indicated MITF or its O-GlcNAc mutants from plasmids were harvested 48 h after transfection, followed by co-IP and FLAG-IPs were then immunoblotted for indicated proteins. ($n = 3$ independent experiments). **c** Identification of the O-GlcNAcylation modification of MITF by in vitro O-GlcNAcylation assays. Recombinant GST-MITF, GST-MITF-S49A, and His-OGT were purified from *E. coli*. ($n = 3$ independent experiments). **d** MCF-7 PR cells were transfected with indicated plasmids for 48 h before being harvested for co-IP. ($n = 3$ independent experiments). FLAG-IPs were then immunoblotted for indicated proteins. **e** Immunofluorescence to examine the MITF localization with indicated treatment in MCF-7 PR cells ($n = 3$ independent experiments). The scale bar represents 10 μm. Right, the quantification of results is shown on the left. **f** MCF-

7 PR cells were transfected with indicated plasmids for 48 h, then cytoplasmic and nuclear fractions were extracted and subjected to immunoblotting for indicated proteins. ($n = 3$ independent experiments). **g** MCF-7 PR cells were transfected with the indicated plasmids for 48 h, and then the nuclear fraction of cell lysates was mixed with biotin-tagged MITF DNA-binding motif, followed by streptavidin pull-down. Streptavidin pull-down was then immunoblotted for indicated proteins. ($n = 3$ independent experiments). **h** MCF-7 PR cells were transfected with indicated plasmids after depletion of endogenous MITF by shRNA, followed by qPCR to examine the expression of indicated genes ($n = 3$ independent experiments). **i** Cell viability was examined in MCF-7 PR cells with indicated treatments ($n = 3$ independent experiments). **j, k** Representative images (**j**) and growth curves (**k**) of MCF-7 PR xenograft tumors with indicated treatments for 3 weeks. $n = 6$ mice/group. $**p \leq 0.01$. All error bars are expressed as mean ± SEM. Two-tailed Student's *t* tests were employed for statistical evaluation. Source data are provided as a Source Data file.

capture disease heterogeneity and, therefore, were widely used as a clinical model for drug discovery and precision oncology[54,55]. We developed a palbociclib resistant organoid from palbociclib resistant PDX WHIM37AR. Using this model, we tested the efficacy of

combined ML329 and palbociclib. As shown in Fig. 7f, ML329 exhibited great synergy with palbociclib in this palbociclib-resistant organoid model. Taken together, these data from clinical samples strongly demonstrate that MITF is up-regulated in palbociclib-

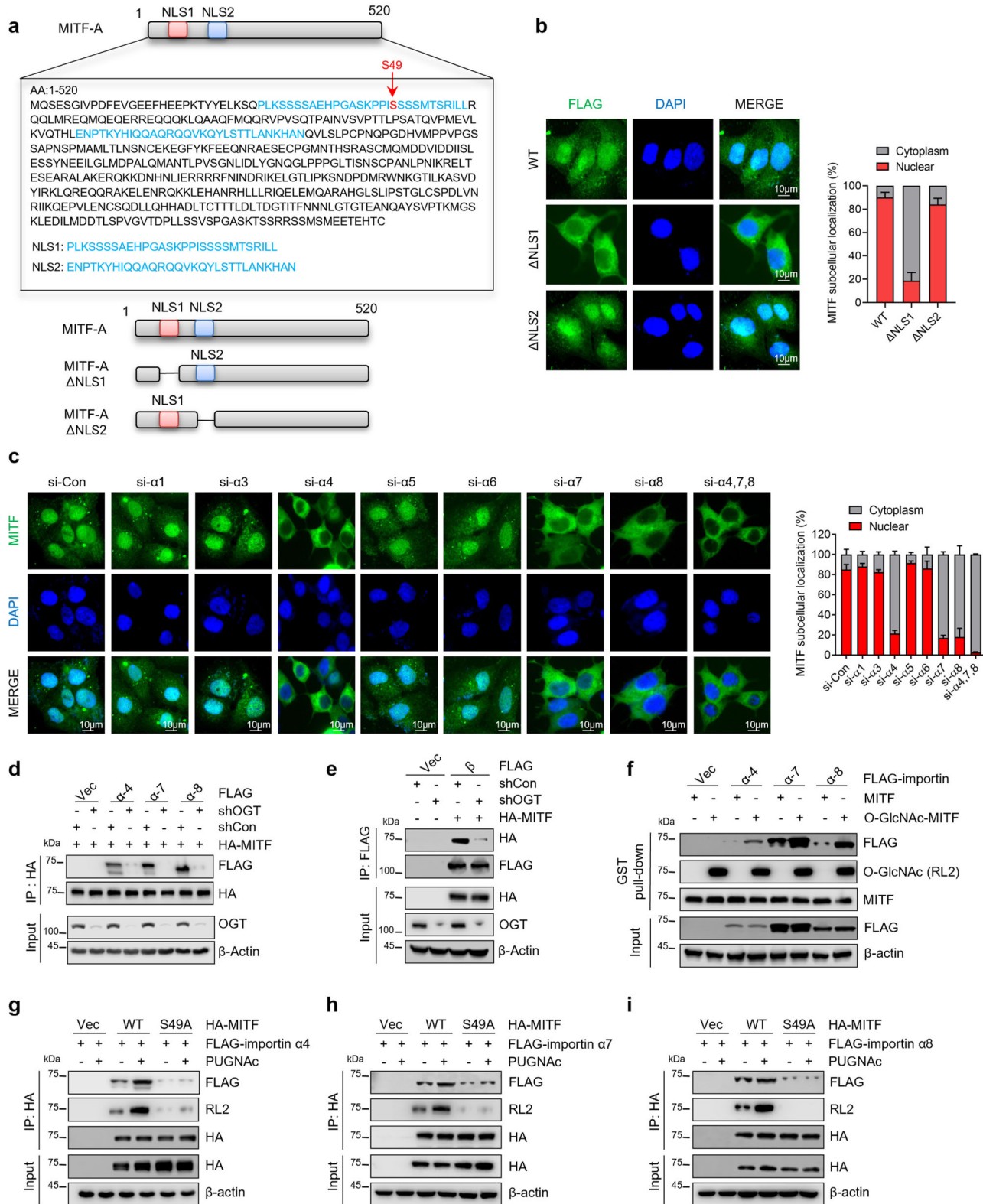

resistant tumors and ML329 could overcome palbociclib resistance in the clinical model.

Given that MITF mRNA is increased in MCF-7 PR cells and palbociclib-resistant patients as compared to their sensitive counterparts, and inhibition of OGT did not affect MITF mRNA level (Supplementary Fig. 6j), we next investigated how MITF mRNA is regulated in resistant cells. After analyzing patient data from the NeoPalAna clinical trial, we found a dramatic increase in the CREB pathway in patients

who received palbociclib treatment (Fig. 7g). Since CREB directly regulates the transcription of MITF[56,57], we assumed that the CREB pathway may regulate MITF mRNA expression in resistant cells. Indeed, the CREB pathway was enriched in palbociclib-resistant cell lines (Fig. 7h, i and Supplementary Fig. 6k). Additionally, palbociclib treatment resulted in a dose-dependent increase of p-CREB expression, and inhibition of the CREB pathway by using its specific inhibitor 666-15 dramatically decreased MITF expression and re-sensitized

**Fig. 5 | O-GlcNAcylation of the nuclear localization signal (NLS) promotes the interaction of MITF with importin α/β and nuclear translocation. a** Amino acid sequence of the MITF putative nuclear localization signals (NLSs). Locations of putative NLS1 and NLS2 were highlighted in blue, and MITF O-GlcNAcylation sites were highlighted in red. The NLS was analyzed using http://nls-mapper.iab.keio.ac. jp/. **b** Endogenous MITF was depleted in MCF-7 PR cells, followed by transfected with indicated plasmids, and MITF cellular localization was examined by immunostaining (*n* = 3 independent experiments). The scale bar represents 10 μm. Right, the quantification results are shown on the left. **c** MCF-7 PR cells were transfected with indicated siRNAs, and MITF cellular localization was detected by immunostaining (*n* = 3 independent experiments). The scale bar represents 10 μm. Right, the quantification results are shown on the left. **d** MCF-7 PR cells transfected with the indicated plasmids were harvested for co-IP. HA-IP was then resolved by SDS-PAGE

and immunoblotted for indicated proteins. (*n* = 3 independent experiments). **e** MCF-7 PR cells transfected with the indicated plasmids were harvested for co-IP. HA-IP was then resolved by SDS-PAGE and immunoblotted for indicated proteins. (*n* = 3 independent experiments). **f** GST-MITF with O-GlcNAcylation was generated by in vitro glycosylation assay. MCF-7 PR cells were transfected with the indicated importin α or vector, and cell lysates were then incubated with GST-MITF with or without O-GlcNAcylation. GST pull-downs were then immunoblotted for indicated proteins. (*n* = 3 independent experiments). **g**–**i** MCF-7 PR cells transfected with the indicated plasmids were harvested for co-IP. HA-IP was then resolved by SDS-PAGE and immunoblotted for indicated proteins. The interactions of MITF with importin α4 (**g**), α7 (**h**), and α8 (**i**) were examined. (*n* = 3 independent experiments). All error bars are expressed as mean ± SEM. Two-tailed Student's *t*-tests were employed for statistical evaluation. Source data are provided as a Source Data file.

resistant cells to palbociclib (Fig.7j–l and Supplementary Fig. 6l, m). Thus, our cell-based studies and clinical evidence indicate that the CREB-dependent pathway plays an important role in the regulation of MITF-mediated palbociclib resistance in breast cancer cells.

## Discussion

In this study, we have discovered a OGT-MITF axis that plays a crucial role in regulating MITF transcriptional activity and conferring resistance to CDK4/6 inhibitors in breast cancer. Specifically, we find that OGT-mediated O-GlcNAcylation of MITF at the S49 site within its nuclear localization signal (NLS) has a dual impact: it enhances the interaction between MITF and importin α/β, while concurrently impeding its association with 14-3-3. This intricate interplay results in the nuclear localization of MITF. In resistant cells, the increased MITF contributes to palbociclib resistance by suppressing senescence through both O-GlcNAcylation and CREB-dependent mechanisms (Fig. 7m). Notably, our study furnishes compelling clinical substantiation of the significance of the MITF in response to palbociclib treatment. We bolster our findings with evidence from PDX lines and patients' tumors, further underlining the clinical relevance of this pathway. This study not only unveils the mechanism governing resistance to CDK4/6i but also offers a potential avenue for the future treatment of palbociclib-resistant breast cancer patients.

The identification of GlcNAcylation in regulating MITF activity holds significant implications. Prior research has demonstrated that MITF activity can be influenced by SUMOylation at two lysine residues, K182 and K316, which is believed to contribute to distinct target specificity[20,58]. Additionally, MITF subcellular localization is regulated via serine phosphorylation by mTORC1, resulting in its interaction with 14-3-3 and cytoplasmic sequestration[59,60]. Our data found that the interaction of MITF-A with 14-3-3 was decreased in palbociclib-resistant cells with higher expression levels of OGT compared to their sensitive counterparts (Supplementary Fig. 3a, b). Inhibition of OGT significantly increased the interaction between MITF and 14-3-3, whereas this interaction was reduced upon PUGNAc (an OGA inhibitor) treatment (Fig. 3h-j). These results indicated that O-GlcNAcylation of MITF-A at its N-terminal NLS exhibits crosstalk with mTORC1-mediated phosphorylation of MITF-A in regulating MITF nuclear localization, which is also consistent with previous evidence that O-GlcNAcylation has extensive crosstalk with phosphorylation[61]. At this moment, we don't know the detailed molecular mechanism of how the O-GlcNAcylation of MITF-A affects its phosphorylation. Most likely, O-GlcNAcylation may change its structure, which further affects its interaction with kinases as well as other proteins. Further investigations aimed at unraveling the coordination between O-GlcNAcylation and the mTOR pathway, particularly at the structural level, could offer insights into the mechanism governing the development of palbociclib resistance in breast cancer cells. Such studies could shed light on intricate regulatory networks and provide avenues for therapeutic intervention.

Importin α functions as an adaptor protein, recognizing and binding to cargo proteins containing NLSs, while importin β acts as a mediator, facilitating the transport of the importin α/cargo-protein complex through the nuclear pore complex (NPC) and into the nucleus[62]. Our previous findings demonstrated that O-GlcNAcylation at or near cargo proteins' NLS region promotes the interaction of cargo proteins with importin α, thereby promoting the nuclear translocation of cargo proteins[45]. To support this notion, our current study underscored the role of OGT-mediated O-GlcNAcylation in regulating MITF. For example, O-GlcNAcylation enhances the association of MITF with importin α4, α7, and α8, and the MITF S49 mutant displays reduced affinity for these importin proteins. Notably, S49 emerges as a pivotal residue within MITF's initial NLS region, playing a critical role in regulating its nuclear localization. Thus, our data suggest that importin α functions as a reader for O-GlcNAcylated NLSs of MITF, thereby exerting control over its nuclear translocation and superseding cytoplasmic retention of MITF in resistant cells. O-GlcNAcylation of MITF at S49 disrupts its interaction with the 14-3-3 protein while simultaneously bolstering its association with importin proteins, facilitating nuclear translocation. This phenomenon hints at the potential for O-GlcNAcylation to induce conformational changes in MITF structure. In the future, structural analyses of MITF-A with/without O-GlcNAcylation are expected to elucidate the detailed mechanism of how this modification affects its structure and its interactions with other proteins.

MITF serves as a critical mediator of senescence in melanocytes and myeloma cells, primarily by orchestrating processes like cell cycle regulation, DNA repair, oxidative stress response, and the production of SASP factors. This is achieved through its modulation of downstream elements including SERPINE1, IL-6, and CCL2[63]. Despite its substantial role in senescence, the regulation of MITF itself in this context remains poorly understood. Our ChIP-seq analysis has unveiled the direct binding of MITF to promoter regions of key SASP-related genes such as SERPINE1, IL-6, CSF1, and PTGER2, signifying its role in governing their expression. Notably, our study underscores the essential role of O-GlcNAc-modified MITF in the suppression of senescence, as it negatively regulates the expression of SASP genes (Fig. 6). Given that palbociclib induces cell cycle arrest and senescence as a primary mechanism to impede cell growth, we postulate that O-GlcNAc-MITF assumes a crucial role in modulating palbociclib resistance through senescence in breast cancer patients undergoing CDK4/6i therapy. The significance of O-GlcNAc-MITF in contributing to the development of resistance to CDK4/6i through senescence warrants further validation through rigorous clinical investigations. By shedding light on the intricate interplay between O-GlcNAc-MITF and palbociclib response, these studies have the potential to enhance our understanding of resistance mechanisms and pave the way for more effective therapeutic strategies.

Our investigation involving clinical samples has revealed a noteworthy observation – the administration of palbociclib leads to an upregulation in the expression of MITF and its associated gene set in

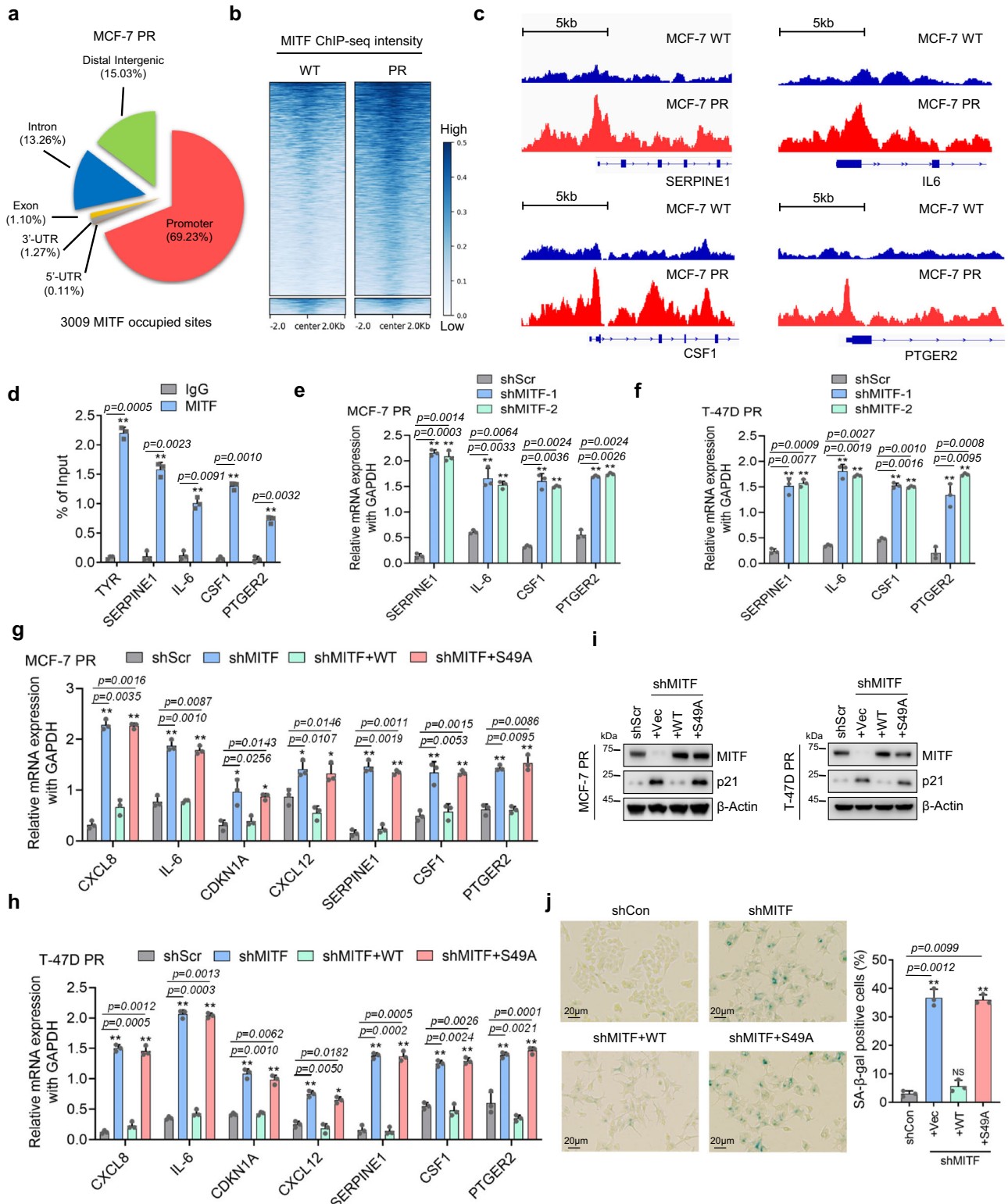

**Fig. 6 | O-GlyNAcylation of MITF contributes to palbociclib resistance by regulating senescence signaling. a** The pie chart shows the distribution of MITF-associated sites across the genomic features in MCF-7 PR cells. **b** The clustering and distribution of MITF-associated sites on genomic DNA around the transcriptional starting sites in MCF-7 PR and MCF-7 cells. **c** ChIP-seq to reveal the association of MITF at the promoter regions of the SERPINE1, IL-6, CSF1, and PTGER2 genes using IGV software. **d** ChIP-qPCR was used to validate the results shown in **c**, TYR was used as a positive control (*n* = 3 independent experiments). **e**, **f** MCF-7 PR cells (**e**) and T-47D PR cells (**f**) were collected after treatments as indicated and subjected to qPCR to examine the expression of genes as indicated (*n* = 3 independent

experiments). **g**–**h** MCF-7 PR cells (**g**) and T-47D PR cells (**h**) were collected after treatments as indicated and subjected to qPCR to examine the expression of genes as indicated (*n* = 3 independent experiments). **i** MCF-7 PR and T-47D PR cells were collected after treatments as indicated, followed by immunoblotting for the indicated proteins. (*n* = 3 independent experiments). **j** SA-β-gal staining of MCF-7 PR cells treated as indicated (*n* = 3 independent experiments). The scale bar represents 20 μm. Right, quantification of results shown on left. **\****p* ≤ 0.01, **\****p* ≤ 0.05. All error bars are expressed as mean ± SEM. Two-tailed Student's *t* tests were employed for statistical evaluation. Source data are provided as a Source Data file.

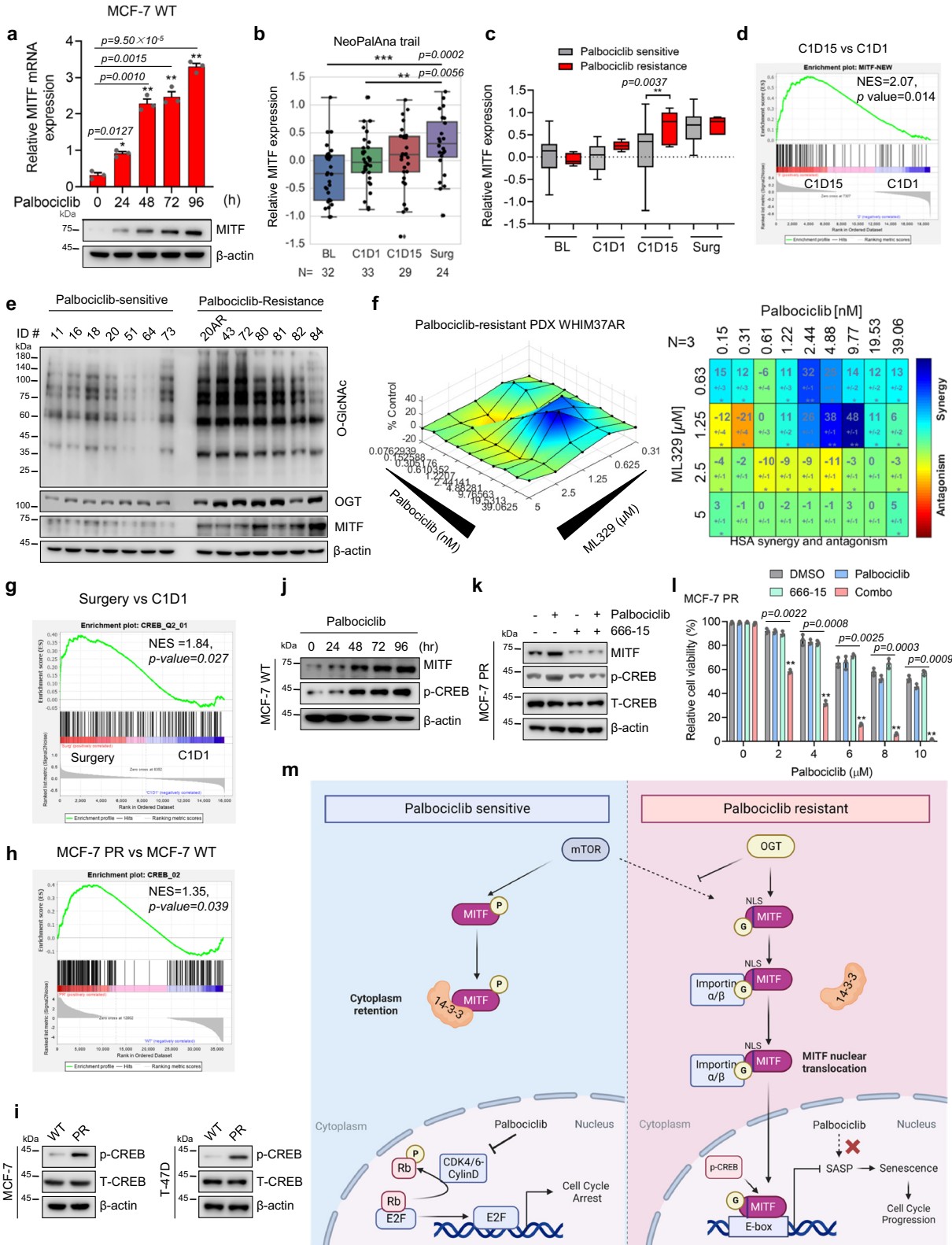

breast cancer patients (Fig. 7). This significant finding suggests that CDK4/6i treatment triggers the elevated expression of the MITF, which may potentially serve as a survival mechanism employed by cancer cells in response to CDK4/6 inhibition. This notion aligns with recent research that highlights elevated MITF expression in basal breast cancer patients, wherein its abundance is linked to unfavorable prognosis[64]. Our clinical studies contribute crucial evidence to support the proposition that MITF is pivotal in orchestrating resistance to

CDK4/6i among breast cancer patients. A particularly compelling instance involves the significantly increased MITF expression in the palbociclib-resistance patients at C1D15 stage compared with sensitive patients (Fig. 7c), and inhibition of MITF exhibited great synergy with palbociclib in the palbociclib-resistant PDX organoid model (Fig. 7f). This patient-specific data is both exhilarating and promising, which indicated that the elevated MITF expression is crucial for palbociclib resistance in breast cancer. In light of these findings, we have

**Fig. 7 | MITF is activated in response to palbociclib and is elevated in tumors from palbociclib-resistant breast cancer patients. a** MCF-7 cells were collected after indicated treatments and then subjected to qPCR and immunoblotting to examine expression levels of genes as indicated ($n = 3$ independent experiments). **b** Expression of MITF in primary breast cancers from patients at stages of baseline (BL), 1 month after endocrine treatment (C1D1), 3 months after endocrine and palbociclib treatment (C1D15), and 12 weeks after endocrine and palbociclib treatment (Surgery) in the NeoPalAna trial ($n$ represents the number of patient biopsies, noted in each graph). Box plots with centerline = median, box = 25th–75th percentile, and whiskers = 5th–95th percentile, outliers = open circles, tests are two-tailed. **c** Boxplots of MITF expression in palbociclib-sensitive and resistant patient groups at indicated stages. Box plots with centerline = median, box = 25th–75th percentile, and whiskers = 5th–95th percentile, outliers = open circles, tests are two-tailed. **d** GSEA profiling to show the enrichment of the MITF signature geneset in the C1D15 group vs. the C1D1 group. NES score and $p$ values determined by GSEA software. **e** Palbociclib-resistant and sensitive breast cancer PDX lines were collected and subjected to immunoblotting for proteins as indicated. **f** Surface plots (left) and synergy matrix (right) to exhibit the synergy between palbociclib and ML329 in palbociclib-resistant PDX WHIM37AR ($n = 3$ independent experiments). **g** GSEA profiling to show the enrichment of the CREB geneset in the Surgery group vs. C1D1 group. **h** GSEA profiling to show the enrichment of CREB geneset in MCF-7 PR cells vs. MCF-7 cells. **i** MCF-7 and T-47D cells were collected and then subjected to immunoblotting for indicated proteins. **j, k** MCF-7 PR cells were collected and then subjected to immunoblotting for indicated proteins. ($n = 3$ independent experiments). **l** Cell viability was examined in MCF-7 PR cells treated with chemicals as indicated ($n = 3$ independent experiments). **m** Schematic of how O-GlcNAcylation of MITF contributes palbociclib resistance in breast cancer cells. See text for details. **$p \leq 0.01$, *$p \leq 0.05$. All error bars are expressed as mean ± SEM. Two-tailed Student's $t$ tests were employed for statistical evaluation. Source data are provided as a Source Data file.

embarked on a comprehensive clinical study encompassing a substantial cohort of patients receiving CDK4/6i treatment. This ongoing endeavor is aimed at amassing more robust clinical evidence to fortify our discovery. The ultimate aspiration is to propel this revelation towards potential clinical applications, thereby advancing our understanding of resistant mechanisms and fostering avenues for therapeutic intervention.

## Methods

### Ethics statement
All experiments described in this study comply with all relevant ethical regulations. Experiments involving animals adhered to a protocol with approval from the Institutional Animal Care and Use Committee (IACUC) at George Washington University.

### Cell culture
Human breast cancer cell lines MCF-7 (HTB-22) and T-47D (HTB-133) were purchased from ATCC and cultured in DMEM medium supplemented with 10% FBS at 37 °C with 5% $CO_2$. Cells were routinely tested for Mycoplasma. The palbociclib-resistant cell lines MCF-7 PR and T-47D PR were generated by exposing them to palbociclib with gradually increasing concentration over 6 months. Only early-passage (<10 passages) resistance cells were used for the study.

### Antibodies and plasmids
The antibodies MITF (CST #12590, 1:1000), phospho-Rb (Ser 780) (CST #9307, 1:1000), phospho-Rb (Ser 807/811) (CST #8516, 1:1000), total-Rb (CST #9309, 1:1000), p21 (CST #2947, 1:1000), β-Actin (CST #3700, 1:1000), PARP (CST #9532, 1:1000), OGT (CST #24083, 1:1000), His-tag (CST #12698, 1:1000), FLAG-tag (CST #14793, 1:1000), Myc-tag (CST #2276, 1:1000), HA-tag (CST #3724, 1:1000), 14-3-3 (pan) (CST #95422, 1:1000), Lamin B1 (CST #13435, 1:1000), Phospho-CREB (Ser 133) (CST #9198, 1:1000), total-CREB (CST #9197, 1:1000) were from Cell signaling technology; antibody against O-GlcNAc (RL2) (ab2739, 1:1000) was from Abcam. MITF-A was amplified through PCR and cloned to pcDNA3.1-FLAG, pCMV-HA, PIRES-FLAG. Plasmid expressing Myc-tagged OGT, FLAG-importin α1- α7, and FLAG-importin β was a kind gift from Dr. Huadong Pei. MITF mutations were generated by site-directed mutagenesis according to the manufacturer's instructions. For generating the stable MITF knock-down cells, two independent shRNAs for MITF were cloned to pLKO.1 vector. The relevant targeting sequences were as follows: shMITF-1, CCAACTTCTTTCATCAGGAAA; shMITF-2, GCCAGACTTGTATATTCTATT.

### siRNA transfections and PCR assay
The following sequences are siRNA duplexes used in this study: siOGT-1, GAUUAAGCCUGUUGAAGUC; siOGT-2, GCUUGCAAUUCAUCACUUU; si-importin-α1, UCAUGUAGCUGAGACAUAA; si-importin-α3, GCCCUCUCUUACCUUACUG; si-importin-α4, UUGUCCUCCACAA ACAUAU; si-importin-α5, GCCUUUGAUCUUAUUGAGC; si-importin-α6, CUAUGCUUGAAAGUCCUAU; si-importin-α7, CGGAGAAAUGUGG AGCUGA; si-importin-α8, UCAGAUCCAGUCCUAUGUU; si-importin-β, CGGAGAUCGAAGACUAACA. All transfections were performed with Lipofectamine RNAi MAX (Invitrogen) according to the manufacturer's instructions. Cells were harvested for analyses 48 h after the siRNA transfection. Total RNA was extracted using RNeasy Mini Kit (OMEGA) and cDNA was prepared using FastQuant RT Kit (TIANGEN), and gene expression was analyzed by real-time PCR. All primers were listed in the Supplementary Data 1.

### Quantitative high-throughput combinatorial screen
Compound screen experiments were performed in two rounds as described previously[65]. MCF-7 and MCF-7 PR cells were used against two compound libraries including MIPE (Mechanism Interrogation Plate) and NPC (NIH Pharmaceutical Collection) library. Briefly, MCF-7 and MCF-7 PR cells (1000 cells/well) were plated in 1536 wells and incubated for 24 h. In the first screen, drugs were tested at 8 different concentrations from 0.8 nmol/L to 46 µmol/L by serial dilution (1:3). After incubation for 72 h, a total of 120 compounds that efficiently inhibited the proliferation of both MCF-7 and MCF-7 PR cells were identified. To further identify compounds that have synergy with palbociclib in MCF-7 PR cells, the top 20 compounds were selected for the second screen together with palbociclib. These 20 compounds were screened at 10 different concentrations in combination with 10 different concentrations of palbociclib. Then 4-µL/well ATP content cell viability assay reagent (Promega, G7570) was added into each well and incubated for 15 min followed by detection of cell viability. Through the second screen, compounds that exhibited the increased cytotoxicity of palbociclib in MCF-7 PR cells were identified. The primary screen data and curve fitting were analyzed using software developed at the NIH Chemical Genomics Center (NCGC).

### RNA-seq analysis
Total RNA from cells MCF-7, MCF-7 PR, and MCF-7 PR transfected with control or MITF shRNA was extracted and prepared for cDNA libraries within Illumina TruSeq Stranded mRNA Sample Preparation Kit (Illumina catalog no. RS-122-2103). Gene-set enrichment analysis (GSEA) was performed by using GSEA software and Hallmark signatures as suggested. RNA-Seq data was listed in Supplementary Data 2.

### Chromatin immunoprecipitation and sequencing
MCF-7 and MCF-7 PR cells were collected and subjected to chromatin immunoprecipitation using kit (SimpleChIP Enzymatic Chromatin IP kit, CST #9003) according to the manufacturer's instructions. Briefly, cells were cross-linked with the 0:4% paraformaldehyde, then rotated for 10 min at RT before quenching with glycine to a final concentration of 0.2 M for 15 min. After washing, cells were lysed with ChIP lysis buffer, and sonicated for approximately 5 min. The sonicated

chromatin was incubated with MITF antibody and rotated overnight. The ChIP-Grade Protein G Magnetic Beads were then added, followed by rotation for 2 h. The beads were washed and eluted with elution buffer. Reverse cross-linking of ChIPed-DNA was performed at 55°C overnight with the addition of 0.3 M NaCl, 20 mg RNase A, and 20 mg Proteinase K. Recovery of ChIPed-DNA was performed by using QIAquick PCR Purification Kit (QIAGEN; Cat# 28106). The concentration of ChIPped-DNA was assessed using Qubit dsDNA HS Assay Kit (Invitrogen; Cat# Q32851). All samples were subjected to sequencing using a HiSeq 4000 (Illumina) at the GENEWIZ genomic service.

### Immunofluorescence assay and sulforhodamine B assay

The immunofluorescence assay was performed as we described previously[66]. Briefly, cells were fixed with 4% paraformaldehyde in PBS for 15 min. Cells were then washed and incubated for 10 min in blocking buffer (PBS containing 3% BSA and 0.02% Tween-20) and subsequently incubated for a primary antibody at room temperature and then incubated with a secondary antibody (rabbit Alexa Fluor-594 and mouse Alexa Fluor-488 were from Life Technology). After washing cells were mounted with Fluoromount G (SouthernBiotech) containing DAPI. The sulforhodamine B assay was performed as described previously[67]. Briefly, cells were fixed with 10% (wt/vol) trichloroacetic acid and stained for 30 min, after which the excess dye is removed by washing repeatedly with 1% (vol/vol) acetic acid. The protein-bound dye is dissolved in 10 mM Tris base solution for OD determination at 510 nm using a microplate reader.

### Senescence-associated β-galactosidase assay

The senescence-associated β-galactosidase staining was conducted in MCF-7 PR cells with the kit (Cell signaling technology, #9860) according to the manufacturer's instructions.

### Biotin pull-down assay

The oligonucleotides containing MITF binding motif CATGTG region were used for Biotin-pull down assays. Briefly, cells were lysed and nuclear fraction was collected, and incubated with biotin-labeled oligos and 20 μl of streptavidin beads overnight. Beads were then washed three times and subjected to SDS-PAGE and blotted with antibodies as indicated.

### Co-immunoprecipitation

Cells were lysed with Co-IP buffer containing 20 mM Tris–HCl at pH8.0, 100 mM NaCl, 1 mM EDTA, 0.5% NP-40, 10 mM NaF, and protease inhibitor cocktail (Roche) on ice for 30 min, followed by sonication 10 sec for three times. After centrifuge, the supernatant was collected and incubated with protein A/G beads coupled with the antibody against indicated proteins at 4°C overnight. The beads were washed three times and analyzed by Western blot. For FLAG IP, cell lysates were incubated with Anti-FLAG-M2 Affinity beads overnight and IPs were analyzed by Western blot.

### Mass spectrometry

MCF-7 PR cells were transfected with FLAG-vector ($n = 2$) or FLAG-MITF ($n = 2$) for 48 h, then cells were harvested for immune precipitation (IP). Eluted proteins after immunoprecipitation were then resolved by SDS-PAGE followed by Commassie blue staining. Corresponding gel bands were excised and in-gel digestion with trypsin, with the digests analyzed by a nanoUPLC-MS/MS system as previously described[68]. In brief, peptides were analyzed with nanoAcquity UPLC coupled with an Orbitrap Lumos mass spectrometer in the EThcD mode. The MS raw data were processed using the Proteome Discoverer platform (version 2.5, Thermo Scientific) and Sequest HT algorithm. the customized MITF protein database containing the two isoforms (i.e., isoform A1 + isoform A2) of human MITF (uniprot) was used for database search.

Two missed tryptic cleavages were allowed, and the minimum peptide length was set to seven amino acids. Variable modifications included oxidation(M) and HexNAc (S/T). Fixed modification included carbamidomethylation (C). MS and MS/MS ion tolerances were set at 10 ppm and 0.02 Da, respectively. The false-discovery rate (FDR) was estimated using the fixed value PSM validation.

Manual confirmation was performed for the potential O-GlcNAc sites assigned. Mass Data was listed in Supplementary Data 3.

### In vitro O-GlcNAcylation assays

Recombinant His-OGT protein (kinase doman) (1 mg) was incubated with 2 mg of recombinant GST-MITF or its mutant GST-MITF-S49A in 60 μL reaction volume (50 mM Tris-HCl, 12.5 mM MgCl2, 2 mM UDP-GlcNAc 1 mM DTT, pH 7.5) at 37°C for 4 h. Samples were separated by SDS-PAGE and subjected to immunoblotting with the indicated antibodies.

### GST pull-down assays

GST-MITF fusion protein was purified from *E. coli* and immobilized on glutathione Sepharose 4B columns (GE Healthcare). HEK293T cells transfected with the indicated plasmids were lysed in NETN buffer with protease inhibitor cocktail (Sigma-Aldrich, USA) and incubated with Sepharose beads immobilized with the indicated GST-tagged proteins at 4 °C overnight. After washing the beads were boiled in 60 μL 2×SDS loading buffer and subjected to immunoblotting with the indicated antibodies.

### Animal experiments

Six-week-old female BALB/c athymic nude mice were purchased from Jackson Laboratory. $4 \times 10^6$ MCF-7 PR or MCF-7 PR MITF-depleted cells were subcutaneously injected into the dorsal flank of each mouse. When the tumor volume reached about 100 mm³, all mice were randomized into indicated experiment groups. Palbociclib was administrated intraperitoneally at 25 mg/kg every 2 days for 2 weeks. Saline was used as the control. The maximum tumor size should not exceed a diameter length of 2 cm (20 mm) in any one dimension. Animals with ulcerated tumors should be treated with analgesics or euthanized immediately. Relative tumor volumes were calculated with the formula: $V = A \times (B^2)/2$, where A and B represent the length and width of each tumor, respectively. For immunohistochemistry (IHC) analysis, tumor samples were sliced and conducted standard IHC staining procedures, and DAB staining was done utilizing SuperPicture™ Polymer Detection Kit (Thermo Fisher Scientific) according to the manufacturer's instructions. All animal studies were carried out by the National Institutes of Health regulation concerning the care and use of experimental animals and with the approval of the Institutional Animal Care and Use Committees of George Washington University.

### Identification and analysis of MITF signature geneset in the NeoPalAna trial

MITF signature geneset was created by overlapping proteins that interact with MITF (mass-spec analysis) and genes that are known to be associated with MITF function by PathwayNet[69], and then overlap genes of these two groups were selected and named as MITF signature geneset (Supplementary Data 3). MITF signature geneset was then used as bait to examine the microarray gene expression data obtained from the tumor biopsies in the NeoPalAna trial[53]. Log₂-normalized expression data from baseline samples were used for the analysis. The gene set enrichment analysis (GSEA) software was used for calculating enrichment scores.

### PDX-derived organoids

PDX organoids were generated as previously described[54,55]. Briefly, single cells prepared from PDX tumors were suspended in 50% to 70% growth factor reduced Matrigel (Corning #35623) in breast organoid

media modified with 10% Noggin/Rpondin1 conditioned media and 5 μM Y-27632. Cultures were fed 2-3 times per week and passaged every 7 to 10 days. Organoids were digested using 1 U/mL dispase (STEMCELL Technologies, cat# 07923) for 96-well and 384-well drug IC50 and drug combination assays. Organoid cells were incubated 2–4 days before the addition of drugs. Organoid cells were exposed to drugs for 5 days for the combination assays. Cell viability was quantified using Cell Titer Glo 3D (Promega cat #G9683).

## Statistics and reproducibility
Experiments were repeated independently by indicated biological replicates in the figure legends. Statistical analyses were performed using GraphPad Prism 9.0. Data were represented as mean ± SEM as indicated in the figure legends. Statistical analysis was performed using Two-tailed Student's t-tests or one-way ANOVA. Differences were considered statistically significant at a p-value of less than 0.05. The sample size was estimated based on the variations and mean values.

## Reporting summary
Further information on research design is available in the Nature Portfolio Reporting Summary linked to this article.

## Data availability
Raw RNA-seq data and ChIP-seq data have been deposited in the Gene Expression Omnibus (GEO) database and are associated with the accession number GSE 234514 and GSE 234515. The DNA microarray data of the NeoPalAna dataset is associated with the accession number GSE93204 as previously described[53]. Mass spectrometry data have been deposited in the ProteomeXchange via the PRIDE partner repository and are associated with the accession number PXD042763 and PXD042838. All remaining data can be found in the Article, Supplementary, and Source Data files. Source data are provided with this paper.

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

## Acknowledgements

This work was partially supported by funding from the National Institutes of Health (CA275904, CA258357, and CA247684 to W.Z.), and a grant from the McCormick Genomic and Proteomic Center. Mass spectrometry analysis conducted in this study was in part supported by NCI-P30 CA051008. We thank Dr. Rong Li for constructive reading and editing. Figures 1a and 7m, and Supplementary Fig. 6c are created with BioRender.com, released under a Creative Commons Attribution-NonCommercial-NoDerivs 4.0 International license.

## Author contributions

Y.Z. performed most of the experiments with assistance from S.Z., Y.K., C.P., M.J., Z.L., B.J., X.Z., and J.M. designed and analyzed mass spectrometry data. M.H., Y.Z., and M.S. designed and performed the high-throughput drug screening assay. S.L. and C.M. provided the clinical patient samples. W.Z. supervised the project, co-wrote the manuscript, provided experimental advice, and obtained funding to support the project.

## Competing interests

The authors declare no competing interests

## Additional information

Shunqiang Li or Wenge Zhu.

