## [Peer Review File · Nature Communications]

Reviewers' Comments:

Reviewer #1:

Remarks to the Author:

Zhang et al. show that in CDK4/6 inhibitor resistant cancer cells lines, MITF activity drives resistance. Furthermore, the authors show that MITF is O-GlcNAcylated at Serine 49 which drives MITF import into the nucleus via importin β . MITF inhibition restored palbociclib (CDK4/6 inhibitor). Importantly, MITF activity is increased in tumors of patients undergoing palbociclib. Finally, the authors link the increase in MITF expression with increased activation of the transcription factor CREB. This is a solid well-controlled and rigorous study. The data is spot on, so I just have a couple tweaks to the English.

Text: Try to reduce the use of adjectives in the text such as remarkable. It's hard to quantify an adjective.

Reviewer #2:

Remarks to the Author:

In the manuscript "O-GlcNAcylation of MITF regulates its activity and CDK4/6 inhibitor resistance in breast cancer", Yi Zhang et al. show that palbociclib-resistant breast cancer cells exhibit high levels of MITF-A and its O-GlcNAcylation, which facilitates its translocation to the nucleus where it suppresses palbociclib-induced senescence and, thus, driving resistance. The authors further show that inhibiting MITF re-sensitized resistant breast cancer cells to palbociclib in vitro and in vivo. Finally, the authors observe activation of MITF in PDXs and tumors from patients treated with CDK4/6i.

This is a nice manuscript that identifies an innovative regulatory mechanism of resistance towards CDK4/6i, provides in-depth mechanistic analyses of MITF activation in cell lines resistant to CDK4/6i and shows strong preclinical data in cell lines and xenografts supporting therapeutic approaches targeting MITF to treat CDK4/6i resistant breast cancer patients. However, there is a lack of clinical data supporting the authors' preclinical findings and the PDX models included should have been used in parallel with the cell line xenografts. These weaknesses substantially diminish the enthusiasm for this manuscript. See below my suggestions to address these shortcomings.

The authors show that MITF levels are induced after treatment with CDK4/6i both in vitro and in vivo and in samples from a neoadjuvant clinical trial. However, it is important to investigate how MITF levels correlate with treatment response, particularly in the advanced setting. Are the tumors exhibiting higher levels of MITF upon treatment correlating with shorter response to CDK4/6i (advanced) or less shrinkage of tumors (neoadjuvant)? This information is critical to support the role of MITF in resistance to CDK4/6i.

The authors investigate the effect of a MITF inhibitor (ML329) in breast cancer cells resistant to CDK4/6i and xenografts models with these cells. The authors should use the PDX models sensitive and resistant to CDK4/6i (included in Fig 7.) to test the effect of MITF inhibition on tumor volume. These are more clinically relevant models and should be used for the mechanistic investigations throughout the manuscript.

Reviewer #3:

Remarks to the Author:

In this manuscript, Zhang et al. show that resistance to Palbo is dependent on MITF-A and that the

resistance is mediated through interactions with OGT which OglcNAcylates MITF at S409 thus affecting nuclear import and transcription, resulting in repression of senescence. The paper is clearly written and the figures nicely display the results. The authors have clearly identified MITF as an important mediator of Palbo-resistance in breast cancer cells and tumors. There are, however, several questions regarding the effects of OGT in particular. First, why does OGT inhibition and mutation of S49 lead to effects on both nuclear localization and DNA-binding? The particular modification mediated by OGT on S49 does not affect the DNA-binding domain of MITF (common to all isoforms). Rather, it affects a domain in an alternative 5'-exon of MITF that is quite distant from the DNA-binding domain. Also, it is not clear how this domain/modification affects interactions with 14-3-3 which have been shown to take place with MITF-M which lacks this amino-terminal domain. Also, the major effects of the S49A mutation on nuclear transport needs further explanation. A major nuclear localization domain is present in the DNA-binding domain of MITF so it is rather surprising to see the clear effects on nuclear transport when they mutate one of their two presumed nuclear localization domains at the 5'-end of MITF. It is not at all clear how the authors came to the conclusion that the two presumptive nuclear localization domains might actually be NLS domains. Does this sequence fit importin-interacting domains? Indeed, since the S49A mutant is dead in all the assays performed, I would suggest that the authors re-analyze this mutant to make sure that it is OK (e.g. does it affect protein stability? Is the sequence consistent with what was expected). Furthermore, the ChIP-seq results comparing wild-type to resistant cells is strange as it does not show peaks but rather broad domains of binding which are not characteristic for MITF and need to be explained. Why would the interacting genes form a signature as shown in Figure 7C? This needs better justification and seems picked at random. Also it is not clear what the 'MITF-mediated pathway' is that is discussed on page 17. Why is the expression of OGT increased in the resistant melanoma cells? Is it a target of MITF? In melanoma cells, MITF usually runs as a double band where the upper band is phosphorylated at S73 (of MITF-M). However, in this manuscript (e.g. Fig 3C) they show only a single band. Is this a cell-specific difference? In the product notes to this antibody (CST #12590) it says that it may cross-react with the related TFE3 protein so extra care should be taken in interpreting the results.

Reviewer #1 - O-GlcNAC (Remarks to the Author):

Zhang et al. show that in CDK4/6 inhibitor resistant cancer cells lines, MITF activity drives resistance. Furthermore, the authors show that MITF is O-GlcNAcylated at Serine 49 which drives MITF import into the nucleus via importin β . MITF inhibition restored palbociclib (CDK4/6 inhibitor). Importantly, MITF activity is increased in tumors of patients undergoing palbociclib. Finally, the authors link the increase in MITF expression with increased activation of the transcription factor CREB. This is a solid well-controlled and rigorous study. The data is spot on, so I just have a couple tweaks to the English.

We thank the reviewer for overall positive comments on our manuscript.

(1) Text: Try to reduce the use of adjectives in the text such as remarkable. It's hard to quantify an adjective.

Thanks for the comments. We have corrected them as suggested.

Reviewer #2 - Breast cancer precision oncology, CDK4/6 inhibitors (Remarks to the Author):

In the manuscript "O-GlcNAcylation of MITF regulates its activity and CDK4/6 inhibitor resistance in breast cancer", Yi Zhang et al. show that palbociclib-resistant breast cancer cells exhibit high levels of MITF-A and its O-GlcNAcylation, which facilitates its translocation to the nucleus where it suppresses palbociclib-induced senescence and, thus, driving resistance. The authors further show that inhibiting MITF re-sensitized resistant breast cancer cells to palbociclib in vitro and in vivo. Finally, the authors observe activation of MITF in PDXs and tumors from patients treated with CDK4/6i.

This is a nice manuscript that identifies an innovative regulatory mechanism of resistance towards CDK4/6i, provides in-depth mechanistic analyses of MITF activation in cell lines resistant to CDK4/6i and shows strong preclinical data in cell lines and xenografts supporting therapeutic approaches targeting MITF to treat CDK4/6i resistant breast cancer patients. However, there is a lack of clinical data supporting the authors' preclinical findings and the PDX models included should have been used in parallel with the cell line xenografts. These weaknesses substantially diminish the enthusiasm for this manuscript. See below my suggestions to address these shortcomings.

We thank the reviewer for overall positive comments on our manuscript. We have addressed all concerns as below.

(1) The authors show that MITF levels are induced after treatment with CDK4/6i both in vitro and in vivo and in samples from a neoadjuvant clinical trial. However, it is important to investigate how MITF levels correlate with treatment response, particularly in the advanced setting. Are the tumors exhibiting higher levels of MITF upon treatment correlating with shorter response to CDK4/6i (advanced) or less shrinkage of tumors (neoadjuvant)? This information is critical to support the role of MITF in resistance to CDK4/6i.

Thanks for the comments. We re-analyzed the neoadjuvant data and found that MITF expression level is significantly elevated in the palbociclib-resistant patients compared to sensitive patients at the C1D15 stage after palbociclib treatment. The evidence that the tumors exhibited higher levels of MITF compared to sensitive patients upon palbociclib treatment in resistant patients suggests that MITF plays an important role in the regulation of CDK4/6i resistance in breast cancer. The new data are now included in Figure 7C.

(2) The authors investigate the effect of a MITF inhibitor (ML329) in breast cancer cells resistant to CDK4/6i and xenografts models with these cells. The authors should use the PDX models sensitive and resistant to CDK4/6i (included in Fig 7.) to test the effect of MITF inhibition on tumor volume. These are more clinically relevant models and should be used for the mechanistic investigations throughout the manuscript.

Thanks for the comments. The PDX samples we tested in Fig. 7E were from PDX lines kept in liquid nitrogen and currently we don't have PDX lines growing in mice. Since ER+ breast PDX lines grow very slowly in vivo, we expect it will take at least one year to test the efficacy of combined treatment using these PDX lines in vivo. To address this concern, we used an alternative clinical model-organoid to test the efficacy of combined treatment. Breast cancer organoids capture disease heterogeneity and therefore

were widely used for drug discovery and precision oncology (Sachs et al., Cell, 2018, 172:373; Guillen et al., Nat Cancer, 2022, 3:232). We have successfully developed palbociclib resistant ER+ breast cancer organoids called WHIM37AR. ML329 exhibited great synergy with palbociclib in palbociclib resistant organoids. This new data was included in Fig. 7F.

Reviewer #3 - MITF signaling (Remarks to the Author):

In this manuscript, Zhang et al. show that resistance to Palbo is dependent on MITF-A and that the resistance is mediated through interactions with OGT which O-GlcNAcylates MITF at S409 thus affecting nuclear import and transcription, resulting in repression of senescence. The paper is clearly written and the figures nicely display the results. The authors have clearly identified MITF as an important mediator of Palbo-resistance in breast cancer cells and tumors. There are, however, several questions regarding the effects of OGT in particular.

We thank the reviewer for overall positive comments on our manuscript. We have addressed the reviewer's questions below.

(1) First, why does OGT inhibition and mutation of S49 lead to effects on both nuclear localization and DNA-binding? The particular modification mediated by OGT on S49 does not affect the DNA-binding domain of MITF (common to all isoforms). Rather, it affects a domain in an alternative 5'-exon of MITF that is quite distant from the DNA-binding domain.

Our study focuses on MITF-A (not MITF-M). As we addressed in the introduction part of manuscript, unlike MITF-M which is constitutively retained within nuclei, whereas MITF-A localizes in the cytoplasm and features an N-terminal IB1b domain, which is absent in MITF-M. We therefore assumed that the Exon1B1b domain may play a key role in regulating MITF-A's cytoplasmic localization. Indeed, our study indicates that O-GlcNAcylation of MITF-A at S49 within the IB1b domain leads to its recognition by importin α , resulting in its nuclear localization. This discovery is consistent with our previous study, in which importin α was found to act as a reader of O-GlcNAcylated NLSs of cargo proteins (SRPK2, RELA, and Sp1) for their nuclear translocation (Tan et al, Molecular Cell, 2021, 6;81:1890-1904.e7).

Inhibition of OGT or mutation of the S49 site significantly reduces the interactions of MITF-A with importins $\alpha 4$, $\alpha 7$, $\alpha 8$, as well as importin β , therefore preventing its nuclear accumulation. Since MITF-A S49 mutants mainly localize in cytoplasm, it is not surprising that MITF-A S49 mutant's DNA-binding activity is compromised (see Figure 3K, Figure 4G, and Figures 5D-5I). Thus, OGT and O-GlcNAcylation of MITF-A at S49 regulate its DNA binding activity via controlling its nuclear localization.

(2) Also, it is not clear how this domain/modification affects interactions with 14-3-3 which have been shown to take place with MITF-M which lacks this amino-terminal domain.

Thanks for the comments. Our data showed that in palbociclib-resistant cells, MITF-A translocation from cytoplasm to nuclei is dependent on its O-GlcNAcylation by OGT at S49 within IB1b domain, which is missing in MITF-M. O-GlcNAcylation of MITF-A increases the interaction of MITF-A with importins for its nuclear translocation, but reduces the interaction between MITF-A and 14-3-3 (Figures 4-5). At this moment, we are not clear on the detailed molecular mechanism regulating these interactions. We believe that in the future, structural analyses, such as the co-crystal structure of MITF-A with 14-3-3 are expected to elucidate the detailed mechanism of how this modification affects the interaction of MITF-A with 14-3-3. We have added this discussion in the Discussion part.

(3) Also, the major effects of the S49A mutation on nuclear transport needs further explanation.

Thanks for the comments. Our previous study has demonstrated that importin α is a reader of O-GlcNAcylated NLSs of cargo proteins and O-GlcNAcylation promotes nuclear translocation of these cargo proteins (Tan et al, Molecular Cell, 2021, 6;81:1890-1904.e7). In our current study, we identified O-GlcNAcylated NLS at the N-terminus of MITF-A (Figure 5A), and found that S49A mutation significantly reduced MITF-A nuclear accumulation (Figures 4E-4F) as well as the associations of MITF-A with importin $\alpha 4$, $\alpha 7$ and $\alpha 8$ (Figures 5G-5I). These results collectively suggest a new mechanism regulating the nuclear translocation of MITF-A via O-GlcNAcylated NLS. This study further supports the notion that O-GlcNAcylated NLSs of cargo proteins are an important mechanism regulating the cellular localization of cargo proteins.

To clarify this mechanism, we have revised our manuscript on page 20 lines 538-547.

(4) A major nuclear localization domain is present in the DNA-binding domain of MITF so it is rather surprising to see the clear effects on nuclear transport when they mutate one of their two presumed nuclear localization domains at the 5'-end of MITF. It is not at all clear how the authors came to the conclusion that the two presumptive nuclear localization domains might actually be NLS domains.

Thanks for these comments. It is true that MITF-M has an NLS close to the DNA-binding domain. However, this study focuses on MITF-A (not MITF-M). In the introduction part of the manuscript, we have summarized the significant differences between these two proteins, including cellular localization (MITF-A is localized in the cytoplasm, but MITF-M is mainly localized in nuclei) and IB1b domain is absent in MITF-M. These differences imply the divergent regulatory pathway controlling the nuclear localization of MITF-A in breast cancer cells compared to MITF-M in melanoma cells. One of the significant differences between MITF-A and MITF-M is that MITF-A has a IB1b domain at its N-terminus, which is absent in MITF-M, indicating that this IB1b domain, as well as its modification, may play a major role in the regulation of MITF-A nuclear localization. Indeed, our study discovered that the O-GlcNAcylation of an NLS at the IB1b domain of MITF-A regulates its nuclear localization.

Two putative NLS within the MITF-A were obtained via the prediction program at <http://nls-mapper.iab.keio.ac.jp/> (Kosugi et al., (2009) PNAS 106, 10171-10176). The best way to determine whether or not a putative NLS is a real NLS is to test whether depletion or mutation of this NLS impairs nuclear location. Our study indicated that deletion of the first NLS but not the second one compromised its nuclear translocation (Figures 5A-5B), indicating that the first NLS (S49 is localized in the first NLS) is a bona fide NLS that regulates MITF-A nuclear localization. It is not surprising to see that MITF-A has a different NLS compared to MITF-M to regulate its nuclear localization given that both proteins exhibit completely different nuclear localization patterns.

(5) Does this sequence fit importin-interacting domains? Indeed, since the S49A mutant is dead in all the assays performed, I would suggest that the authors re-analyze this mutant to make sure that it is OK (e.g. does it affect protein stability? Is the sequence consistent with what was expected).

Thanks for the comments. We have mapped the domain of MITF-A (T1:1-220aa, T2: 221-370aa, T3: 371-520aa) (**Figure 1 A**) and found that the N-terminal T1 domain (containing NLS) but not T2 and T3, is responsible for the interaction with the importin $\alpha 4$, $\alpha 7$, and $\alpha 8$. (**Figure 1 B-D**).

Additionally, inhibition of OGT doesn't affect MITF protein expression level, only regulates MITF nuclear accumulation ability (see Supp Figure 4D).

Figure 1. The associations of MITF-A and its truncated mutants with importin $\alpha 4$, $\alpha 7$, and $\alpha 8$. **(A)** Schematic of the MITF-A protein domains and truncated mutants (T1, T2, and T3) used for protein-protein interactions. **(B-D)** HA-MITF-A and its mutants, as well as FLAG-importin $\alpha 4$ **(B)**, $\alpha 7$ **(C)**, and $\alpha 8$ **(D)**, were expressed in 293T cells. HA-IPs were resolved by SDS-PAGE and immunoblotted for the indicated proteins.

(6) Furthermore, the ChIP-seq results comparing wild-type to resistant cells is strange as it does not show peaks but rather broad domains of binding which are not characteristic for MITF and need to be explained.

Thanks for the comments. This is because of scale we used for ChIP-seq is too large, therefore reducing the size of peaks. We re-analyzed the data using the new scale and the new data was included in Figure 6C. Clearly, MITF displayed a stronger binding affinity in the promoter region of SERPINE1, IL-6, CSF1, and PTGER2 in resistance cells compared to sensitive cells.

(7) Why would the interacting genes form a signature as shown in Figure 7C? This needs better justification and seems picked at random.

Thanks for the comments. We define the MITF signature gene by considering not only MITF physical interacting proteins but also known functionally MITF-associated genes. We believe that physical and

functional-related genes with MITF play a crucial role in the regulation of resistance mediated by MITF. The same strategy has been widely used in the field to define signature gene sets as described previously (e.g. Dominik Saul, *Nat Commun*, 2022, 13:4827; Minjae Yoo, *Bioinformatics*, 2015, 31:3069-71; Ricardo Avila, *Nucleic Acids Res* 2023,51:350-356; Tânia Barata, *Genes (Basel)*, 2023, 14:745)

(8) Also it is not clear what the 'MITF-mediated pathway' is that is discussed on page 17.

Thanks for the comments. To avoid confusion, we have changed "MITF-mediated pathway" to "MITF" in the manuscript text.

(9) Why is the expression of OGT increased in the resistant melanoma cells? Is it a target of MITF?

Thanks for the comments. We assume that the reviewer asked for "resistant breast cancer cells" not "melanoma cells". We have conducted the experiments as suggested. The results indicate that depletion of MITF does not affect the level of OGT mRNA and protein (**Figure 2 A and B**). These data suggest that OGT is not a target of MITF.

Figure 2, (A) MCF-7 PR cells were collected after treatment as indicated and subjected to qPCR and Western to examine the expression of genes and proteins as indicated. (B) T-47D PR cells were collected after treatment as indicated and subjected to qPCR and Western to examine the expression of genes and proteins as indicated. **, $p \leq 0.01$.

To investigate how OGT expression is regulated in palbociclib-resistance cells, we examined the OGT mRNA level and found the increased expression of OGT in resistance cells compared with sensitive cells (**Figure 3A**). A recent study indicated that p300 regulates OGT transcription activation through C/EBP β (J Biol Chem, 2018, 293:13989-14000). We then examined the EP300 and CEBPB mRNA levels and found that both genes were increased in resistant cells. Meanwhile, depletion of EP300 or CEBPB by siRNA significantly reduced OGT mRNA expression level (**Figure 3B**), while co-depletion of EP300 and CEBPB further reduced OGT mRNA level (**Figure 3C**). Thus, elevated OGT expression was mainly regulated by EP300 and CEBPB in palbociclib-resistant cells.

Figure 3. (A) MCF-7 WT & PR cells and T-47D WT & PR cells were collected and subjected to qPCR to examine the expression of genes as indicated. (B) MCF-7 WT & PR cells and T-47D WT & PR cells were collected and subjected to qPCR to examine the expression of genes as indicated. (C) MCF-7 PR cells and T-47D PR cells were collected after treatment as indicated and subjected to qPCR to examine the expression of genes as indicated. *,

(10) In melanoma cells, MITF usually runs as a double band where the upper band is phosphorylated at S73 (of MITF-M). However, in this manuscript (e.g. Fig 3C) they show only a single band. Is this a cell-specific difference?

Thanks for the comments. In this study, the isoform of MITF is MITF-A, not MITF-M (Supp Fig. 2A). Moreover, in this study, we used ER+ breast cancer cells, not melanoma cells. Most likely, these differences may explain why only a single band for MITF is seen in our study.

(11) In the product notes to this antibody (CST #12590) it says that it may cross-react with the related TFE3 protein so extra care should be taken in interpreting the results.

Thanks for the comments. We have performed the experiments as suggested. The results (see below) showed that the MITF antibody (CST #12590) does not cross-react with TFE3 (CST #81744) in our cells.

Note: MITF antibody does not recognize the TFE3 protein, which displays a different molecular weight from MITF in the Western blotting. The specificity of both antibodies was confirmed by shRNA treatments.

Reviewers' Comments:

Reviewer #2:

Remarks to the Author:

The authors have satisfactorily addressed my comments.

Reviewer #3:

Remarks to the Author:

In this revision the authors address some of my concerns but ignore some of my other comments. They seem to misunderstand the difference between the A and M isoforms of MITF. The A isoform contains everything that the M isoform contains, except for the specific M-exon at the 5'-end of MITF which encodes only 11 amino acids. In addition, the A isoform contains the exon 1B1b and the 1A exon, encoding a total of 138 residues that are not present in M (see Vu et al., 2020). Thus, the A isoform contains the strong nuclear localization signal present in the basic domain of MITF as well as a nuclear export signal around S73 of the M isoform (S180 in the A isoform)(Ngeow et al., 2018). The M-isoform has been shown to interact with 14-3-3 and this interaction involves S173 of M (S280 in the A isoform). Here they show a presumed new nuclear localization signal in the A isoform that is not present in the M-isoform. Does this new nuclear localization signal override the other signals? How does that override mechanism work? It should be noted that the supposed NLS they have identified partly overlaps with the domain affected by the TOR pathway (reference 26). Furthermore, there is a Degron sequence and RAG-binding domains in the A-specific isoform of MITF that are known to have a major effect on structure and stability of this domain (Nardone et al., 2023). The authors need to show that the S49 residue and the NLS around it confer (i) a nuclear localization signal to a heterologous protein and (ii) that this heterologous protein now mediates interactions with OGT/importins.

Surprisingly, the authors maintain their claim that the S49A mutation affects DNA binding of MITF-A. Unfortunately, they can not claim this based on the experiment they did. They isolated nuclei from cells expressing MITF before and after OGT knockout and then used the nuclear extracts to test DNA binding. However, it is clear that OGT knockouts lead to cytoplasmic retention of MITF-A so there is no MITF in the nucleus under those conditions. Absence of MITF from the nucleus is the only conclusion to be made here and no claims can be made with respect to DNA binding. The S49A mutation does not affect the very well characterized DNA binding domain of MITF and would be unlikely to affect DNA-binding ability.

Reviewer #3 (Remarks to the Author):

In this revision the authors address some of my concerns but ignore some of my other comments. They seem to misunderstand the difference between the A and M isoforms of MITF. The A isoform contains everything that the M isoform contains, except for the specific M-exon at the 5'-end of MITF which encodes only 11 amino acids. In addition, the A isoform contains the exon 1B1b and the 1A exon, encoding a total of 138 residues that are not present in M (see Vu et al., 2020).

We thank the reviewer for the comments and apologize for misunderstanding some comments raised by the reviewer on last revision. We have addressed reviewer's all questions as below.

(1) Thus, the A isoform contains the strong nuclear localization signal present in the basic domain of MITF as well as a nuclear export signal around S73 of the M isoform (S180 in the A isoform)(Ngeow et al., 2018). The M-isoform has been shown to interact with 14-3-3 and this interaction involves S173 of M (S280 in the A isoform). Here they show a presumed new nuclear localization signal in the A isoform that is not present in the M-isoform. Does this new nuclear localization signal override the other signals? How does that override mechanism work? It should be noted that the supposed NLS they have identified partly overlaps with the domain affected by the TOR pathway (reference 26).

Thanks for these comments.

*It was reported that Ser73 of the M isoform (Ser180 in the A isoform) is required for MITF's nuclear export signal transduction (Ngeow et al, PNAS, 2018). Therefore we investigated whether OGT-mediated O-GlcNAcylation of MITF-A exhibits crosstalk with phosphorylation of MITF-A at Ser 180. As shown in **Figure 1A** as below, knockdown of OGT significantly increased p-MITF level in MCF-7 PR cells, similar results were observed in cells treated with OGT inhibitors OSMI-1 or BADGP (**Figure 1B**). These results suggested that O-GlcNAcylation of MITF-A may regulate its nuclear translocation by affecting its phosphorylation at Ser 180, a known mechanism regulating nuclear localization of MITF (Ngeow et al, PNAS, 2018).*

Figure 1. OGT regulates MITF phosphorylation. (A) MCF-7 PR cells were transfected with the indicated siRNAs for 48hr, followed by immunoblotting for the indicated proteins. (B) MCF-7 PR cells were collected after treatments as indicated, followed by immunoblotting for the indicated proteins.

It was also reported that MITF subcellular localization is regulated by serine phosphorylation mediated by mTORC1, resulting in its interaction with 14-3-3 and cytoplasmic sequestration (Kaushal et al, Nat Commun, 2022; MartinaM J.A. et al, Autophagy, 2012; Settembre, C. et al, EMBO J, 2012). Our following data indicate that O-GlcNAcylation of MITF-A also crosstalks with mTORC1/14-3-3-mediated pathway:

*(i) The interaction of MITF-A with 14-3-3 was decreased in palbociclib-resistant cells (higher levels of OGT expression) compared to their sensitive counterparts (See Manuscript **Supl. Figure. S3A-3B**).*

*(ii) OGT depletion or treatment with the OGT inhibitor OSMI-1 significantly increased the interaction between MITF and 14-3-3 (See Manuscript **Fig. 3H-3I**), whereas this interaction was reduced upon PUGNAc (an OGA inhibitor) treatment (See Manuscript **Figure. 3H-3J**).*

In conclusion, our data suggest that O-GlcNAcylation of MITF-A exhibits crosstalk with known mechanism regulating MITF nuclear localization, which is consistent with reviewer's comments. These data is also consistent with evidence that O-GlcNAcylation has extensive crosstalk with phosphorylation (Gerald W. Hart et al, Annu Rev Biochem, 2011; Saar A.M. et al, FEBS J, 2018). These two PTMs may compete or coordinate with each other to regulate signal transductions (Changmin Peng et al, Mol Cell, 2017; Xiaoyan Li et al, Oncogene, 2021; Lorela Ciraku et al, Oncogene, 2022). To clarify this point, we have added extra discussion in the Discussion part.

At this moment, we don't know the detailed molecular mechanism of how O-GlcNAcylation of MITF-A affects its phosphorylation. Most likely, O-GlcNAcylation may change its structure, which further affects its interaction with kinases as well as other proteins. In future, structural analysis of O-GlcNAcylation MITF-A could help to solve this mystery.

(2) Furthermore, there is a Degron sequence and RAG-binding domains in the A-specific isoform of MITF that are known to have a major effect on structure and stability of this domain (Nardone et al., 2023). The authors need to show that the S49 residue and the NLS around it confer (i) a nuclear localization signal to a heterologous protein and (ii) that this heterologous protein now mediates interactions with OGT/importins.

Thanks for these comments. We have done experiments as suggested.

*We tested whether NLS we identified in MITF-A is able to redirect nuclear translocation of a cytoplasmic translation factor-eIF2Bε. To this end, we constructed a plasmid pcDNA3.1-HA-NLS-eIF2Bε, in which NLS was added to the N-terminus of eIF2Bε. We then examined the localization of HA-eIF2Bε or HA-NLS-eIF2Bε in MCF-7 PR cells. The results indicated that HA-eIF2Bε was localized in the cytoplasm as previously reported (Michael Schoof et al, Elife, 2021; Jacquelyn et al, FEBS lett, 2019). However, HA-NLS-eIF2Bε was mainly localized in the nucleus (**Figure 2A shown below**).*

Moreover, as suggested by reviewer, we examined the interaction of HA-NLS-eIF2B ϵ with importin α/β by co-IP. The results indicated that the NLS-eIF2B ϵ exhibited an increased interaction with importin $\alpha4$ and β compared to the eIF2B ϵ (Figure 2B shown below). Thus, NLS we identified is sufficient to mediate nuclear localization and the association with importin α/β .

Figure 2. eIF2B ϵ subcellular localization. (A) pcDNA3.1-HA-eIF2B ϵ and pcDNA3.1-HA-NLS-eIF2B ϵ plasmids were transfected into MCF-7 PR cells as indicated, and the eIF2B ϵ cellular localization was examined by immunostaining. (B) MCF-7 PR cells were transfected with indicated plasmids for 48 hr before harvested for co-IP. HA-IPs were then immunoblotted for indicated proteins.

(3) Surprisingly, the authors maintain their claim that the S49A mutation affects DNA binding of MITF-A. Unfortunately, they can not claim this based on the experiment they did. They isolated nuclei from cells expressing MITF before and after OGT knockout and then used the nuclear extracts to test DNA binding. However, it is clear that OGT knockouts lead to cytoplasmic retention of MITF-A so there is no MITF in the nucleus under those conditions. Absence of MITF from the nucleus is the only conclusion to be made here and no claims can be made with respect to DNA binding. The S49A mutation does not affect the very well characterized DNA binding domain of MITF and would be unlikely to affect DNA-binding ability.

We apologized for misunderstanding the reviewer's previous question. We now addressed this question as below.

We conducted *in vitro* assay to test whether S49A mutation affects DNA binding of MITF-A. Specifically, FLAG-tagged WT MITF-A or S49A MITF-A were expressed from plasmids in MCF-7 PR cells. Cell lysates were then mixed with biotin-tagged oligonucleotide containing the MITF binding motif (E-box)(see detail of this assay in Manuscript **Fig 3K**). The results indicated that FLAG-tagged WT MITF-A and S49A MITF-A showed little difference in affinity with E-box DNA (**Figure 3 as below**).

Thus, the O-GlcNAcylation of S49 in MITF-A mainly regulates its nuclear localization rather than its DNA binding activity. To clarify this point, we have modified the statements in manuscript accordingly.

Figure 3. DNA binding affinity of MITF-A-WT and S49A. MCF-7 PR cells were transfected with indicated plasmids, and the whole cell extracts were mixed with biotin-tagged MITF binding motif (E-box), and then subjected to the biotin pull-down assay, followed by immunoblotting for indicated proteins.

Reviewers' Comments:

Reviewer #3:

Remarks to the Author:

The authors have satisfied my previous concerns. I would however like to request that they include some of the discussion included in reply to my comment 1 in the discussion section of the manuscript and that they add their review figure 2 to the main manuscript. This is an important piece of evidence that they have identified a nuclear localization signal in the N-end of the A-isoform.

Reviewer #3 (Remarks to the Author):

The authors have satisfied my previous concerns. I would however like to request that they include some of the discussion included in reply to my comment 1 in the discussion section of the manuscript and that they add their review figure 2 to the main manuscript. This is an important piece of evidence that they have identified a nuclear localization signal in the N-end of the A-isoform.

We thank the reviewer for the comments. We have revised our manuscript based on the suggestions. The revised parts are in lines 419-421 of page 16 and lines 545-561 of page 20-21.